# *P-Law*: Predicting Quantitative Scaling Law with Entropy Guidance in Large Recommendation Models

**Tingjia Shen**[1], **Hao Wang**[1]*, **Chuhan Wu**[2], **Chin Jin Yao**[2], **Wei Guo**[2],
**Yong Liu**[2], **Huifeng Guo**[2], **Defu Lian**[1], **Ruiming Tang**[2], **Enhong Chen**[1]

[1]State Key Laboratory of Cognitive Intelligence, University of Science and Technology of China
[2]Shenzhen Huawei Technologies Co.Ltd.
 **Source Code:** https://github.com/USTC-StarTeam/P-Law

## Abstract

With the growing size of data and models in Large Recommendation Models, the time required for debugging has become increasingly prohibitive, underscoring the urgent need for effective guidance in parameter configuration. The **Scaling Law** (SL) offers analogous guidance in the Sequential Language domain, having achieved significant success by predicting model loss when scaling model size. However, the existing guidance from SL for Sequential Recommendation (SR) remains qualitative, which is because quantitative analysis of SL on SR encounters challenges with quality measurement on redundant sequences along with loss-performance discrepancy. In response, we introduce the **Performance Law** (*P-Law*) for SR models, which predicts model performance across various settings, intending to provide a quantitative framework for guiding the parameter optimization of future models. Initially, Performance Law utilizes Real Entropy to measure data quality, aiming to remove the low-quality influence of low-entropy redundant sequences. Subsequently, Performance Law investigates a fitting decay term, which facilitated the prediction of the major loss-performance discrepancy phenomena of overfitting, ultimately achieving quantitative performance prediction. Extensive experiment on various datasets demonstrates the effectiveness of Performance Law by displaying exceptional quantitative prediction ability against the original and modified qualitative SL. Additional application experiments on optimal parameter prediction and model expansion potential prediction also demonstrated the broad applicability of the Performance Law.

## 1 Introduction

Recently, billions of data points are generated daily across various application platforms for large recommendation models(1). To effectively model this data, large-scale recommendation models have been introduced due to their potential to enhance performance(2; 3). As larger recommendation models are developed, optimizing the expensive parameters of these models(4) leads to high costs and unpredictable performance during development. This challenge motivates researchers to explore the Scaling Law (SL) to efficiently plan time and GPU consumption across various model sizes(5). The concept of SL was initially investigated within the realm of Large Language Models (LLMs) (6). The introduction of the Chinchilla model (7), which provides a quantitative framework for modeling the final pre-training loss $L(N, D)$ based on the number of model parameters $N$ and the number of tokens $D$, has led to the adoption of this principle in models such as LLaMA2 (8) and Mistral (9).

---

*Correspondence to `wanghao3@ustc.edu.cn`

39th Conference on Neural Information Processing Systems (NeurIPS 2025).

As SL has made progress with LLMs, researchers are interested in extending these laws to large-scale recommendation models, as they share similar Transformer architectures. Several large Sequential Recommendation (SR) models, including HSTU(10) and Wukong(11), have developed qualitative recommendation SL to assess model effectiveness under various scenarios. Research across several domains has similarly reported performance enhancements due to large-scale models, evident in Click-Through Rate (CTR) prediction(12), reranking(13), and retrieval (14). Nonetheless, current studies primarily remain qualitative, highlighting the need for advancing toward a quantitative prediction of model performance.

Our work attempts to extend the Scaling Law to enable quantitative prediction and analysis of SR model performance. However, as illustrated in Figure 1, this approach faces the following two challenges: (1) **Quality Measure Deficiency**. Currently, the analysis of data in SL is measured using token counts. While this is relatively reasonable in language models, SR often exhibits variable vocabulary sizes, redundant user sequences, and random item noise. This results in significant discrepancies in the amount of information contained within the same token count, necessitating the introduction of quality metrics to refine the SL. (2) **Loss-Performance Discrepancy**. While SR systems prioritize model

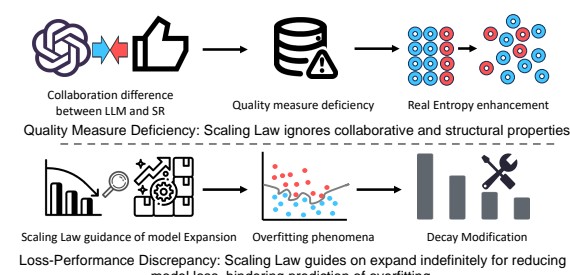

Quality Measure Deficiency: Scaling Law ignores collaborative and structural properties.

Loss-Performance Discrepancy: Scaling Law guides on expand indefinitely for reducing model loss, hindering prediction of overfitting.

Figure 1: Two challenges when expanding the origin Scaling Law to quantitative predict SR performance: Quality Measure Deficiency and Loss-Performance Discrepancy, along with our approach: Real Entropy enhancement and Decay Modification.

performance, SL models focus on analyzing and predicting training loss. However, as overtrained SR models frequently display overfitting (15; 16) behaviors, training loss does not necessarily correspond to actual model performance. Thus, even with the enhancement of Quality Measures to quantitatively analyze model scaling loss, a reduction in model loss does not always translate into significant performance improvements. We term this dilemma the Loss-Performance Discrepancy challenge.

To address these challenges, we introduce a novel **Performance Law** that studies the SR system with an entropy-enhanced performance analysis formula, thereby quantitatively predicting the performance of SR models. Specifically, we first collected model performance data across a wide range of different model parameters for analysis. Subsequently, in response to challenge (1), we define the Data Scale as the minimum encoding length of datasets, aiming to minimize encoding redundancy on specific datasets and thereby enable adaptive research across different encoding vocabularies. Next, we introduce the concept of Real Entropy (17) $S^{real}$ as a novel measure of data quality. By calculating the distributional differences in user interactions using $S^{real}$, we propose a correction for the removal of redundant low-entropy item sequences in SR, ultimately facilitating the transfer of SL from LLMs to SR models. Toward challenge (2), we investigated the consistency between testing loss and two metrics: normalized discounted cumulative gain (NDCG) and hit rate (HR), aiming to address the inconsistency between loss and performance, facilitating quantitative modeling and prediction of model performance. Subsequently, we introduce a decay term into the fitting formula for SL, altering the previously predicted trend of monotonically increasing performance with model size into a rise-then-fall pattern. This modification enables the quantitative prediction of model overfitting phenomena, ultimately establishing the Performance Law for quantitative guidance on model parameters and framework. Our contributions are summarized as follows:

- We developed Performance Law, the first quantitative approach to predict model performance across various settings, intending to provide a more quantitative framework for anticipating the effects of how both data and model parameters change will guide the optimization of future models.

- Performance Law innovatively utilizes minimum encoding length to measure data scale and Real Entropy $S^{real}$ to remove low-quality influence of low-entropy redundant sequences, enhancing the transfer of SL in SR. This approach enables quantitative analysis of data in SR.

- Performance Law further analyzes the difference between training loss and performance, enabling quantitative analysis of model performance. This is accomplished through extending a fitting decay term, which facilitates the quantitative prediction of overfitting.

- Extensive experiment on various datasets demonstrates the effectiveness of Performance Law by displaying exceptional quantitative prediction ability against qualitative SL. Experiments on optimal model parameter prediction and model expansion potential further validate the applicability and promise of the Performance Law.

## 2 Related Work

### 2.1 Sequential Recommendation

The focus of recommendation systems has undergone significant transformations. Among these, sequential recommendation (18) is a technique that aims to delve into and understand users' interest patterns by analyzing their historical interactions (19; 20). Initially, statistical analysis techniques such as Markov chains (21) and collaborate filtering (22) were employed in sequential recommendation (SR). However, with the emergence of neural networks, deep-learning approaches have been developed for sequential recommendation tasks. Convolutional Neural Networks(CNN) (23), Graph Neural Networks (GNN) (24) from GCE-GNN (25) to SR-GNN (26). However, the compatibility of RNN (27) with sequences has ultimately garnered more attention. Early work like GRU4Rec (28) and Caser (29) were introduced to improve recommendation accuracy. Another notable technique in sequential recommendation is the attention mechanism. SASRec (30), for instance, utilizes self-attention to independently learn the impact of each interaction on the target behavior. On the other hand, BERT4Rec (31) incorporates bi-directional transformer layers after conducting pre-training tasks. Since LLaMA4Rec (32) and HSTU (10) demonstrated improvements in SR performance brought by large models and datasets, it is meaningful to study how the model performance would change as the model scales up.

### 2.2 Scaling Law on Large Sequential Models

Scaling laws were first explored in the context of Large Language Models (6; 18). Specifically, since the introduction of the Chinchilla scaling law, which models the final pre-training loss $L(N, D)$ as a function of the number of model parameters $N$ and the number of training tokens $D$, models such as LLaMA2 (8), Mistral (9), and Gemma (33) have applied this principle. Empirical evidence indicates that model performance consistently improves with increased model size and training data volume (6; 34; 35; 36) and in Sequential Recommendation (10; 11; 37; 38; 3). Extensive experiments have explored neural scaling laws under various conditions, including constraints on computational budget (39), data limitations (40), and regeneration (41), and instances of over-training (42). (43) tailoring content to individual preferences towards expanding modules, while (44) further processing has been done to refine SL's description of model precision However, increasing the model size does not necessarily lead to better performance. Some studies have observed a decline in performance due to overfitting (45; 46). Following theoretical analysis, (47) and (48) empirically validated this point, underscoring the necessity for an enhanced understanding of scaling laws. However, as mentioned in the introduction, these methods are still qualitative analyses, while quantitative analyses are facing challenges related to Quality Measure Deficiency and Loss-Performance Discrepancy, resulting in only superficial and non-quantitative analysis for Sequential Recommendation.

## 3 Preliminary and Definition

### 3.1 Problem Definition

**Definition1. (Scaling Law)** Scaling Law indicates that model performance consistently improves with increased model size and training data volume. These analyses employ a decomposition of expected risk, resulting in the fit of $L(N, D) = \left[ \frac{N_C}{N}^{\frac{\alpha_N}{\alpha_D}} + \frac{D_C}{D} \right]^{\alpha_D}$, where L denotes the model Loss, N is the number of parameters, D is the number of tokens in the dataset in SL, $N_C, D_C, \alpha_N$, and $\alpha_D$ are parameters. In some works, the scaling law formula might be simplified to $L'(N, D, \theta) = E + \frac{A}{N^\alpha} + \frac{B}{D^\beta}$ with parameter $\theta = \{\alpha, \beta\}$. The goal of SL is to minimize $|\hat{L} - L'|$ with $\theta$, where $\hat{L}$ represents the actual model training loss, making a more accurate prediction.

**Definition2. (Performance Law)** The only distinction between the Performance Law and the SL is replacing the research target $L(N, D)$ with $P(N, D)$, where $P$ represents the model's performance.

To delve further into model performance, we refine model parameter $N$ into more granular components with the number of layers $H$, the hidden layer dimension $h$, and the embedding dimension $d_{emb}$. To ensure alignment between the input and output for model stacking, we set $h = d_{emb}$. Consequently, the formula we ultimately aim to derive is $P(H, d_{emb}, D, \theta^*)$, where $\theta^*$ denotes the parameter set fitting the Performance Law. In contrast, the objective of the Performance Law is to minimize $|\hat{P} - P'|$ with $\theta^*$, where $\hat{P}$ represents the actual model performance.

## 3.2 Preliminary

**Real Entropy** We introduce the Real Entropy (17) ($S^{real}$) factor to enhance scaling laws in SR further. Real entropy is a refined measure that captures user interaction distributions across patterns of varying lengths. It is computed as: $S^{\text{real}} = -\sum_{T' \subset T} P(T') \log_2[P(T')]$, where $T$ represents the complete sequence formed by concatenating all user interaction sequences in the dataset, $P(T')$ represents the probability of each subset of transitions $T'$. By calculating the entropy of distributional differences in user interactions, we make a correction for removing redundant low-entropy item sequences in SR, ultimately addressing the quality measure deficiency in transferring SL from LLM to SR. However, the complexity of calculating $S^{real}$ using the above definition is extremely high. To reduce this complexity, we utilize the following lemma:

**Lemma 3.1.** *LZ compression (49). For a user interaction sequence with length $|S_u|$, its Real Entropy $S^{real}$ is estimated by:* $S^{real} = \left( \frac{1}{|S_u|} \sum_j \Lambda_j \right)^{-1} \ln |S_u|$, *, where $\Lambda_j$ denotes the minimum length $j$ such that the subsequence starting from position $i$ with length $j$ does not appear as a continuous sub-sequence of $S_u = [i_1, i_2, \ldots, i_{|S_u|}]$.*

From Lemma 3.1, it can be observed that Real Entropy $S^{real}$ has a positive correlation with the model's duplication rate, whereas conventional entropy tends to have a negative correlation with data duplication rate. Therefore, although $S^{real}$ is referred to as entropy, its trend is opposite to traditional entropy. To avoid confusion, we use $S'^{real} = 1/S^{real}$ as the final measure of Real Entropy.

# 4 Methodology

## 4.1 Outline of Performance Law

Overall, we identify two main challenges in applying Scaling Laws: the deficiency of data quality evaluation and the loss-performance discrepancy of model overfitting. To clearly illustrate the entire process, we present the complete pipeline of the Performance Law in Figure 2. Specifically, as shown in Part A, we first collected model performance data across a wide range of different model parameters for analysis. Regarding challenge (1), we measure the data scale $D'$ utilizing the Minimum Encoding Length $C_{min}$ with $D' = |U|C_{min}$, as it represents the minimal token pattern constraint sufficient to encode the entire dataset sequence, accommodating the analytical needs of different vocabularies across various datasets. We then offer a theoretical proof on the lower bound of data scale in Theorem A.2, establishing that the data scale $D'$ is constrained by $D' \geq \#Tokens \cdot S'^{real}$, which is shown on Lemma 3.1 and illustrated in Part B Figure 2. For challenge (2), we incorporate a decay term $\frac{1}{\cdot} + \log(\cdot)$ when the model layer $H$, embedding dimension $d_{emb}$ and data scale $D$ are scaled up, as demonstrated in Theorem A.4 and illustrated in Part C Figure 2. Building upon the theorems outlined, we will complete the construction of the Performance Law.

## 4.2 Scaling Parameters

### 4.2.1 Scaling Model Parameters

We first construct the Scaling Transformer Framework to scale up the model-side parameters: the number of layers $H$ and the embedding dimension $d_{\text{emb}}$, which is shown in Part (A) in Figure 2. Following the prior empirical study HSTU (10), for all experiments, we adopt the decoder-only transformer models as the backbone. Specifically, for each user $u$, items in the user behavior sequence $S_u = [i_1, i_2, \ldots, i_{|S_u|}]$ are firstly encoded into embeddings, forming $\mathbf{e}_u = [\mathbf{e}^{i_1}, \mathbf{e}^{i_2}, \ldots, \mathbf{e}^{i_{|S_u|}}]$ with embedding dimension $|\mathbf{e}^i| = d_{emb}$. After the embedding layer, we stack multiple Transformer decoder blocks. At each layer $l \in \{1, 2, \ldots, H\}$, query $Q$, key $K$, and value $V$ are projected from

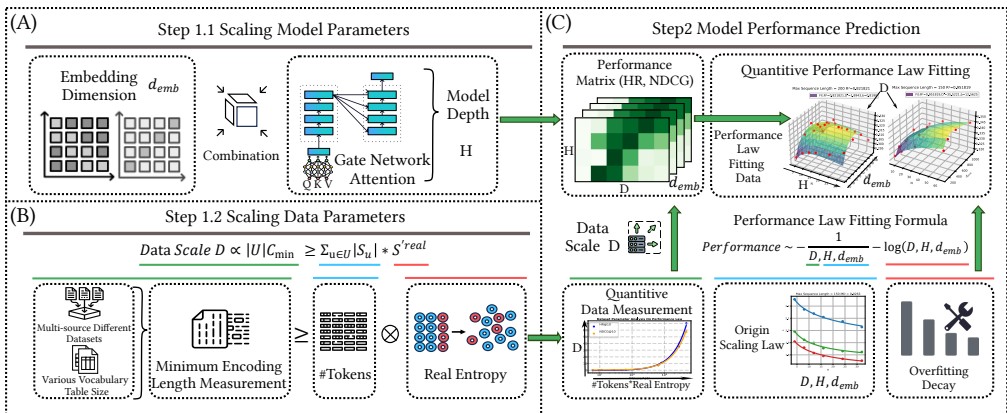

Figure 2: Illustration of Performance Law. Parts A, B, and C in the figure correspond to Section 4.2.1, Section 4.2.2, and Section 4.3 respectively.

the same input hidden representation matrix $R_l$. We modified the $SiLU$ activation module and the $Rab$ positional encoding module within the transformer block to ensure effectiveness. Specifically, the Spatial Aggregation Layer is defined as: $Attn_l(\cdot)V_l = SiLU(Q_lK_l^T + Rab_l)V_l$. Upon deriving matrices $Q$, $K$, and $V$, the spatial aggregation layer utilizes an attention mechanism to adjust value $V$. This layer is distinguished by two key features: the SiLU activation function and a relative attention bias, enriching the model with positional and temporal information, thus enhancing its ability to discern contextual interrelations and capture dependencies within the data. Subsequently, the Pointwise Transformation Layer is defined as follows: $e_{l+1} = Norm(Attn_l(e_l)V_l \odot Gate_l(e_l))$. Following the spatial aggregation layer, the pointwise transformation layer independently transforms each individual data point. Here, the gating weights $Gate_l(\cdot)$ are combined with the normalized values $Norm(Attn_l(\cdot)V_l(\cdot))$ via a Hadamard product, effectively gating the transformed representations and allowing the model to selectively emphasize relevant entries while weakening less significant ones. The results are then transformed by a single-layer MLP. Finally, we then calculate the similarity of this item $\mathbf{e}'^{i_{k+1}}$ with those in the entire item pool $I$, retrieving the most similar item and storing it in the candidate set for each user as $\mathcal{I}^u$. The loss function for the retrieval task is defined as:

$$\mathcal{L}_u = -\log \frac{\exp(\langle \mathbf{e}_H^{i_{|S_u|}}, \mathbf{e}^{i_{|S_u|}} \rangle)}{\exp(\langle \mathbf{e}_H^{i_{|S_u|}}, \mathbf{e}^{i_{|S_u|}} \rangle) + \sum_{\mathbf{v}^- \in Neg_u^-} \exp(\langle \mathbf{e}'^{i_{|S_u|}}, \mathbf{v}^- \rangle)}, \quad (1)$$

where $\langle \cdot, \cdot \rangle$ denotes the dot product, which measures similarity and negative samples $Neg_u^-$ are randomly drawn from the item pool $I \setminus \{i_{|S_u|}\}$, distinguishing the positive item from irrelevant ones.

After constructing the Scaling Transformer model, as shown in Part A, Figure 2, we investigated the impact of varying model parameters on performance by adjusting the number of model layers, $H$, and the embedding dimension, $d_{emb}$. However, beyond model parameters, the scale and quality of the data are also crucial factors influencing performance. In the following sections, we will detail the construction methodology of the data parameter $D$ and its quantitative analysis.

### 4.2.2 Scaling Data Parameters

To fit the Performance Law formula mentioned in Section 3.1, we have collected performance data under different model parameters. However, we still face significant vocabulary differences across datasets and issues of quality measure deficiency. First, we evaluate the Data Scale $D$ using the minimum encoding length $D' = |U|C_{min}$ with user count $|U|$, aiming to minimize encoding redundancy on specific datasets, enabling adaptive research across different encoding vocabularies. We then assess data quality using the Real Entropy $S^{real}$ in Section 3.2. Through our analysis, these two factors exhibit the following relationship:

**Theorem 4.1.** *Assuming that the user sequence can be modeled as a first-order aperiodic stationary Markov chain (50). If the user sequence $S = \{S_u, u \in U\}$, then the relationship between the sum of minimum encoding length $|U|C_{min}$ and Real Entropy $S'^{real}$ is given by:*

$$D \sim D' = |U|C_{min} \geq (\Sigma_u|S_u|) \cdot S'^{real}. \tag{2}$$

The detailed proof of Theorem 4.1 is elaborated in Appendix A.1. At a high level, we establish our results by proving the following two key equations:

$$|U|C_{min} \geq \sum_{\forall S_u} p(S_u) \log_2 \frac{1}{p(S_u)} \geq \frac{1}{ln2} \sum_{\forall S_u} p(S_u) \frac{2(1-p(S_u))}{p(S_u)+1} \geq \frac{\Sigma_u|S_u|(\Sigma_u|S_u|+2)}{4|U| \ln \Sigma_u|S_u|}. \tag{3}$$

$$\frac{1}{S'^{real}} \geq \left(\frac{1}{\Sigma_u|S_u|} \sum_j \Lambda_j\right)^{-1} \ln \Sigma_u|S_u| \geq \left(\frac{2}{|U|\Sigma_u|S_u|} \sum_{j/2} (\frac{\Sigma_u|S_u|}{2} - j)\right)^{-1} \ln \Sigma_u|S_u|$$

$$= \left(\frac{\Sigma_u|S_u|(\Sigma_u|S_u|+2)}{4|U|\Sigma_u|S_u|}\right)^{-1} \ln \Sigma_u|S_u| = \frac{4|U| \ln \Sigma_u|S_u|}{(\Sigma_u|S_u|+2)}, \tag{4}$$

The product of Inequality (3) and Inequality (4) completes the proof of the theorem, while the detailed derivations are provided in Appendix A.1, Inequality (19) and Inequality (17).

Overall, the theory allows us to achieve adaptive adjustments for different vocabularies through minimum encoding length $C_{min}$ and to introduce Real Entropy $S'^{real}$ for analyzing the collaborative distribution across different datasets. This, in turn, enables a more quantitative analysis of the data scale $D = |U|C_{min} = (\Sigma_u|S_u|) \cdot S'^{real}$ in SR models, which is shown in Part (B) in Figure 2. Ultimately, by integrating model depth $H$ and embedding dimension $d_{emb}$ from Section 4.2.1, we obtained model loss and performance across various scaled parameters. This integration allows for a quantitative measurement of Scaling Law $L(H, d_{emb}, D)$ and Performance Law $P(H, d_{emb}, D)$.

### 4.3 Model Performance Prediction

After obtaining the model's loss and performance under different data and model parameters, we need to construct a fitting function for the final model performance. Specifically, we introduce the Squeeze Theorem of Performance Fitting to establish a function for understanding the relationship between model configurations and performance. According to the definition of the Squeeze Theorem, we will identify a reasonable fitting function for model performance evaluation. The whole process is shown in Part (C) in Figure 2. Through our analysis, model performance $P(H, d_{emb}, D)$ is squeezed with the functions below:

**Theorem 4.2.** *Squeeze Theorem of Performance Fitting* There exist $\hat{w}_3, w'_3, \hat{w}_4, w'_4$ such that

$$log(d_{emb}^{\hat{w}_3} D^{\hat{w}_4}) + \frac{1}{d_{emb}^{\hat{w}_3} D^{\hat{w}_4}} + \log H + \frac{1}{H} - \delta \leq P(H, d_{emb}, D)$$

$$\leq log(d_{emb}^{w'_3} D^{w'_4}) + \frac{1}{d_{emb}^{w'_3} D^{w'_4}} + \log H + \frac{1}{H} + \delta, \tag{5}$$

where $\delta$ is a small constant shift. The detailed proof of Theorem4.2 is elaborated in Appendix A.2. At a high level, we establish our results with Lemma A.3 and employing loss-metric consistency:

$$1 < P(H, d_{emb}, D) = \log Z_t + \frac{1}{Z_t} + \log H + \frac{1}{H} - \frac{\Delta_F}{\sqrt{\Delta_\phi}} \cdot \sqrt{\Phi(L_{\text{Metirc}}) - \Phi(L_{test})} - \delta, \tag{6}$$

$$\frac{D \cdot \Psi_{d_{In}}(\sqrt{\frac{d_{In}}{2\pi e}})}{\exp(\sqrt{\frac{d_{In}}{2\pi e}})} \leq Z_t \leq D \cdot \Psi_{d_{In}}(\sqrt{\frac{d_{In}}{2\pi e}}), \tag{7}$$

where $L_{test}$ denotes testing loss, $L_{\text{Metirc}}$ denotes the specific metric (NDCG, HR) loss. $\Psi_{d_{In}}(r) = \pi^{\frac{d_{In}}{2}} r^{d_{In}} / \Gamma(1 + \frac{d_{In}}{2})$, $d_{In} = |S_u|_{max} \times d_{emb}$ is the dimension of input sequence. Building upon the theorems outlined, the model's performance is encapsulated in the following equation:

**Final Formula of Performance Law** $P(H, d_{emb}, D)$:

$$P(H, d_{emb}, D) \sim w_1(\log H^{w_3} + \frac{p_1}{H^{w_3}}) + w_2(\log d_{emb}^{w_4} + \frac{p_1}{d_{emb}^{w_4}}) + \log D + \frac{p_2}{D^{w_5}}.$$

$$D = \Sigma_u |S_u| \cdot S'^{Real} = \#Tokens \cdot S'^{Real}$$

(8)

Ultimately, we will collate the performance matrices obtained by altering various parameters in Section 4.2.1. These matrices will be used to fit the performance model using the specified formula in conjunction with the least squares method. Finally, we will compare the differences with the Scaling Law fitting approach by evaluating their correlation coefficient $R^2$ to determine the quantitative effectiveness and accuracy of our method.

# 5 Experimental Evaluation

**Datasets.** To demonstrate the performance of our proposed approach across various kinds of datasets, we conducted experiments on three publicly available datasets: **MovieLens-1M** (51) (ML-1M), **Amazon Books** (52) (AMZ-Books), **KuaiRand-Pure** (53) (KR-Pure) and one private dataset **Industrial**. The private industrial dataset is drawn from an online music platform that has impressions of more than 400 million users, challenging existing scaling laws. The specific details of the dataset are presented in Table 3 of Appendix A.3.

**Baselines.** Our primary baseline for comparison is the fitting formula of the Scaling Law, which serves as the research paradigm for the majority of SR extension laws. Only the Precision Scaling Law has introduced some modifications. The specific baseline is introduced as follows:

- **Scaling Law** (SL) (54) empirical SL for neural model performance, showing that cross-entropy loss scales with model size, dataset size, and compute, guiding optimal compute budget allocation. The majority of large recommendation models (10; 11; 37; 38; 3) utilize this formula for research.
- **Precision Scaling Law** (PcSL) (44): consider a small shift on low precision towards SL, enabling predictions of loss and guiding compute-optimal strategies by accounting for precision effects on model parameters and degradation.

We divide the comparison with the baseline into two parts to examine the effectiveness of our Quality Measure Extension in Section 5.1.1 and Performance Fitting in Section 5.1.2. Since both baselines measure data by token count, we do not differentiate them in the first part, and will then compare them individually in the second part.

**Experiment Settings.** We adopt the leave-one-out strategy for evaluation, following prior research (55; 56; 57). For each sequence, the most recent interaction is used for testing, the second for validation, and the rest for training. We assess we evaluate the Top-K recommendation performance using HR (58) and NDCG (59). The equipment, time, and specific parameter details used in our model are illustrated in the Appendix A.4.

## 5.1 Overall Validation of Performance Law

### 5.1.1 Experiment Validation on Quality Measure Extension

To verify the rationale for using Real Entropy as a data quality metric, we need to evaluate if it provides a more accurate analysis of data compared to token count $\#Tokens$ alone, as proposed in the original Scaling Law. Specifically, we modified the data coefficient $D$, integrating it as a parameter into both the standard Scaling Law formula (with fitting results shown in Figure 3, $D_{Loss}$) and our enhanced Performance Law formula (with fitting results shown in Figure 3, $D_{HR}$ and $D_{NDCG}$). A higher correlation coefficient $R^2$ indicates a better predictive capability of the data modeling. From the Figure 3, we can draw the following conclusions: (1). Whether in the fitting of data parameters by the Scaling Law (left) or the Performance Law (right), Performance Law with $\#Tokens \cdot S'^{real}$ measurement (blue or green) consistently provides a better fit than $\#Tokens$ in SL (red or orange). This indicates that our proposed Real Entropy successfully enhances data quality, effectively extending the Scaling Law to the more synergistic and structured task of recommendation.

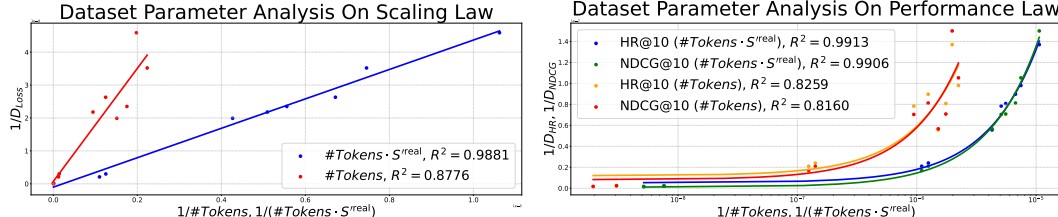

Figure 3: The linear correlation between parameter $D$ and $\#Tokens \cdot S'^{Real}$. The left figure validates this relationship within the context of the SL Loss, while the right figure verifies it within the Performance Law Metric, evaluated with a correlation coefficient $R^2$.

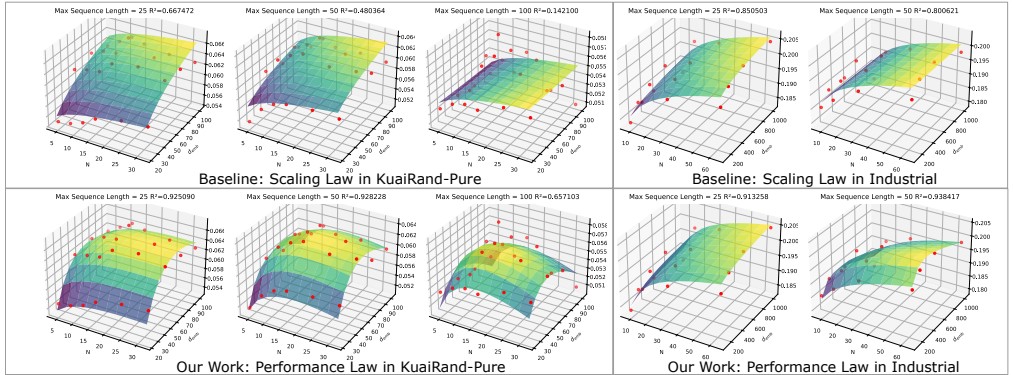

Figure 4: The PL and SL function fitting between model performance (red points) and the prediction (surface) of different kinds of functions: PL and the most authoritative baseline: SL. The plot includes annotations of the correlation coefficients $R^2$. All comparisons are statistically significant with $p < 0.01$. Situations are the same as all other datasets and metrics in Appendix A.6.

(2). $\#\text{Tokens} \cdot S'^{real}$ offers a quantitative foundation on the data level for our subsequent analysis of the Performance Fitting, as both HR and NDCG have a fitting $R^2$ greater than 0.99, which is a relatively high value in the context of the fitting. Additionally, despite the significant differences between these two metrics, the curves obtained from fitting are remarkably similar. This further demonstrates that our enhancement is valid across different metrics of Performance Fitting. The specific numerical values presented in Figure 3 are detailed in Table 3, Appendix A.3.

### 5.1.2 Experiment Validation on Performance Fitting

To verify the resolution of Performance Law on quantitative performance prediction, we fit the model's performance on HR and NDCG and compare among Performance Law, SL, and PcSL with the correlation coefficient $R^2$. The results of Scaling Law Fitting and Performance Law Fitting are depicted in Table 1. To formally demonstrate the improvements of the Performance Law over the Scaling Law, we present the fitting surface of them in Figure 4. By comparing the results of Table 1 between the SL fitting and the Performance Law Fitting, we draw the following conclusions: (1). In smaller datasets (ML-1M, KR-Pure), the Performance Law significantly improves over the Scaling Law. This indicates that our proposed decay terms provide quantitative predictions for model overfitting. In ML-1M, when the maximum sequence length is 200, the improvement can be as much as 345%. (2). In larger datasets (AMZ-Books and Industrial), the Performance Law still demonstrates its strong capability in modeling model performance. This

Table 1: The Performance Law (PL) against SL and PcSL function fitting between model performance with model and data parameters, evaluated with correlation coefficient $R^2$, all results are statistically significant with p<0.01.

| Dataset | Metric $|S_u|$ | $R^2$ (HR@10) | | | $R^2$ (NDCG@10) | | |
|---|---|---|---|---|---|---|---|
| | | **PL** | SL | PcSL | **PL** | SL | PcSL |
| KR-Pure | 25 | **0.792** | 0.671 | 0.672 | **0.925** | 0.667 | 0.667 |
| | 50 | **0.939** | 0.465 | 0.466 | **0.928** | 0.480 | 0.481 |
| | 100 | **0.649** | 0.172 | 0.182 | **0.657** | 0.142 | 0.142 |
| ML-1M | 100 | **0.898** | 0.551 | 0.699 | **0.851** | 0.543 | 0.647 |
| | 150 | **0.916** | 0.764 | 0.756 | **0.882** | 0.740 | 0.735 |
| | 200 | **0.892** | 0.189 | 0.200 | **0.907** | 0.174 | 0.172 |
| AMZ-Books | 25 | **0.892** | 0.803 | 0.837 | **0.904** | 0.855 | 0.879 |
| | 50 | **0.851** | 0.751 | 0.830 | **0.866** | 0.809 | 0.836 |
| Industrial | 25 | **0.939** | 0.855 | 0.856 | **0.913** | 0.850 | 0.851 |
| | 50 | **0.951** | 0.803 | 0.767 | **0.938** | 0.800 | 0.750 |

Table 2: Verification of Performance Law's Optimal Parameter Search Capability. The best parameter marked as "Prediction" in the last row of each table is computed using the Performance Law. All model performance is actual but not predicted. Results are statistically significant with $p<0.05$.

| | H | $d_{emb}$ | NDCG@10 | NDCG@50 | HR@10 | HR@50 | H | $d_{emb}$ | NDCG@10 | NDCG@50 | HR@10 | HR@50 |
|---|---|---|---|---|---|---|---|---|---|---|---|---|
| | | | Global optimal solution | | | | | | Global optimal solution | | | |
| Smallest Dataset ML-1M | 8 | 54 | 0.1831 | 0.2418 | 0.3265 | 0.5916 | 28 | 25 | 0.1732 | 0.2326 | 0.3101 | 0.5791 |
| | 12 | 54 | 0.1824 | 0.2409 | 0.3271 | 0.5913 | 28 | 50 | 0.1866 | 0.2437 | 0.3311 | 0.5917 |
| | 16 | 54 | 0.1853 | 0.2434 | 0.3286 | 0.5903 | 28 | 75 | 0.1810 | 0.2408 | 0.3203 | 0.5882 |
| | 32 | 54 | 0.1810 | 0.2387 | 0.3216 | 0.5837 | 28 | 100 | 0.1726 | 0.2307 | 0.3102 | 0.5741 |
| Prediction | 28 | 54 | **0.1878** | **0.2443** | **0.3322** | **0.5924** | 28 | 54 | **0.1878** | **0.2443** | **0.3322** | **0.5924** |
| | | | Optimal solution (with constraint $H$ =64) | | | | | | Lptimal solution (with constraint $H \cdot d_{emb} \simeq 512$ ) | | | |
| | H | $d_{emb}$ | NDCG@10 | NDCG@50 | HR@10 | HR@50 | H | $d_{emb}$ | NDCG@10 | NDCG@50 | HR@10 | HR@50 |
| Largest Dataset Industrial | 64 | 256 | 0.2019 | 0.2623 | 0.3481 | 0.6205 | 4 | 128 | 0.1758 | 0.2371 | 0.3111 | 0.5854 |
| | 64 | 370 | 0.2035 | 0.2639 | 0.3504 | 0.6226 | 8 | 64 | 0.1773 | 0.2381 | 0.3118 | 0.5858 |
| | 64 | 512 | 0.2032 | 0.2636 | 0.3502 | 0.6226 | 10 | 51 | 0.1758 | 0.2365 | 0.3092 | 0.5840 |
| | 64 | 1024 | 0.1981 | 0.2590 | 0.3448 | 0.6195 | 16 | 32 | 0.1704 | 0.2305 | 0.3007 | 0.5732 |
| Prediction | 64 | 603 | **0.2040** | **0.2644** | **0.3512** | **0.6235** | 12 | 44 | **0.1777** | **0.2383** | **0.3121** | **0.5863** |

success is attributed to our direct analysis of model performance rather than model loss. In all larger datasets, the correlation coefficient $R^2$ exceeds 0.85. Moreover, compared to the Scaling Law, it achieves at least a 2.8% improvement. This also demonstrates that the Performance Law can effectively predict and guide parameter configuration for training as data scale increases. Beyond $R^2$, we also report results using two additional evaluation metrics: Mean Absolute Error (MAE) and Root Mean Squared Error (RMSE) in Appendix D.

## 5.2 Applications of Performance Law

### 5.2.1 Application 1: Global and Local Optimal Parameter Search

A practical application of the Performance Law is to search best parameter by predicting the performance gain from model expansion techniques. We divide the calculation of optimal parameters into two parts: (1) global optimal parameter prediction and (2) optimal parameter prediction under constraints. For the second part, we provide two constraints: one with a constant model depth $H$ and the other with a constant total number of parameters $H \cdot d_{emb}$ (as calculated from (10)). To determine the optimal parameters of the model, we compute the best configuration by fitting the Performance Law and compare it with other parameter configurations. The actual performance in different parameter settings is presented in Table 2, with parameter configuration marked as "Prediction" in the last row of each table computed using the Performance Law. From the results, we draw the following conclusions: (1) The Performance Law exhibits high accuracy. This is evidenced by Performance Law predicted parameters outperforming others in global optimal parameter prediction. This ensures the high potential for application of the guidance on searching optimal parameter of the Performance Law. (2) The Performance Law demonstrates robustness. This is evidenced by Performance Law predicted parameters continuing to outperform others in local optimal parameter predictions under both conditions. This ensures that the Performance Law can yield meaningful guidance on parameter setting in a variety of practical application scenarios.

### 5.2.2 Application 2: Exploring Performance Law Potential Among Framework

Another application of the Performance Law is to assess potential performance gains when scaling up the model. We conducted experiments and fitting analyses on three different frameworks (HSTU (10), LLaMA2 (8), and SASRec (15)) evaluated at different precisions: float32 and bfloat16. Experiments were performed on the smaller dataset (ML-1M) while larger values of $w_1$ and $w_2$ indicate a better scaling-up potential for the model with analysis in Appendix A.5. The fitting results are presented in Table 4 in Appendix A.5. From the table, we conclude that the model's performance closely aligns with the magnitude trend of $w_1$ and $w_2$, further underscoring the accuracy of our quantitative fitting. This also demonstrates that the Performance Law can effectively guide model structure configuration, thereby reducing memory and time when modifying frameworks.

## 5.3 Further Evaluation

After demonstrating the applicability of the Performance Law, we need to further validate our Performance Law across a broader range of models and qualities. We approach the expansion of Performance Law evaluation from the following two aspects

### 5.3.1 Model Extension and Generalization

To further validate the universality and scalability of the proposed *Performance Law*, we extend our experiments to several representative models and diverse recommendation scenarios:

**LightGCN** (60) is a highly efficient graph convolutional network tailored for collaborative filtering. **Mamba** (61) is a recent architecture for sequence modeling leveraging state-space models for high accuracy and scalability. **Wukong** (62) is a stacked factorization machine architecture with a synergistic upscaling strategy, specifically designed to realize scaling laws in recommendation tasks. **DiffuRec** (63) is a novel sequential recommendation framework that leverages diffusion models for item representation construction and uncertainty injection. For long-tail recommendation, we conduct analyses on the challenging Amazon Beauty[2] dataset, where user-item interactions are sparse and item distributions are highly skewed. Complete experimental results and metric comparisons with Scaling Law and Precision Scaling Law baselines for these additional settings are presented in **Appendix B**. Across all models and scenarios, *Performance Law* consistently yields higher $R^2$ and lower error metrics, evidencing its robust generalization and superior predictive precision under both dense and sparse, as well as multimodal recommendation paradigms.

### 5.3.2 Comparison of Data Quality Evaluation Functions

Beyond Real Entropy, we explored a variety of data quality evaluation functions, including Approximate Entropy (64), Shannon Entropy (65), and Kolmogorov Complexity (66). Such alternatives have been widely adopted to measure system complexity and randomness in different domains. To provide a thorough comparison, we conducted experiments by replacing Real Entropy with each candidate function in our framework. As shown in Table 6 in the Appendix C, Real Entropy consistently demonstrates the superior fitting ability over the alternatives. This empirical finding further supports the efficacy of Real Entropy as the preferred choice for characterizing data quality in recommendation scaling law analysis.

## 6 Discussion

**Limitation and Future Directions**. Performance Law has currently been thoroughly tested in the SR domain, but with suitable metrics, our theoretical framework remains applicable to other recommendation tasks. For our future work, we aim to extend Performance Law to larger datasets and a broader range of Recommendation tasks, such as ranking and retrieval.

**Conclusion**. In this paper, we have investigated a novel problem concerning a quantitative approach to predicting SR model performance across various settings. To tackle this problem, we first introduced Performance Law, introducing minimum encoding length and Real Entropy to remove the low-quality influence of low-entropy redundant sequences, providing quantitative data analysis for SL. Subsequently, we further analyzed the difference between training loss and performance by including consistent metrics and a fitting decay term. This facilitated the prediction of overfitting and provided a quantitative analysis of model performance. Performance Law displayed exceptional quantitative prediction accuracy against the original qualitative Scaling Law, successful experiments on optimal model parameter prediction and model expansion potential prediction also demonstrated the broad applicability of the Performance Law.

**Broader Impact**. The Performance Law serves as a framework for quantitatively predicting and analyzing SR performance. While current tasks are limited to recommending items, future applications might overlook fairness issues, leading to potential biases. Nevertheless, the Performance Law can offer effective guidance for the optimal parameter configuration of models.

## 7 Acknowledgement

This work was supported by the National Natural Science Foundation of China (Nos. 62441239, U23A20319, 62472394, and 62441227), as well as the Anhui Province Science and Technology Innovation Project (Nos. 202423k09020010 and 202423k09020011). We also gratefully acknowledge the USTC Supercomputing Center for providing the computational resources essential to this research.

---

[2]https://snap.stanford.edu/data/amazon/productGraph/categoryFiles/

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

## A  Appendix / supplemental material

### A.1  Proof of Theorem 4.1

**Entropy Inequality**  To facilitate our analysis of the Real Entropy, we need to introduce the following entropy inequality:

**Lemma A.1.**  *(67) If $x_i$, $y_i$, $i = 0, 1, ..., n$, and $\Sigma_{i=1}^n x_i = 1 \geq \Sigma_{i=1}^n y_i$, then*

$$\sum_{i=1}^q x_i \log_r \frac{1}{x_i} \leq \sum_{i=1}^q x_i \log_r \frac{1}{y_i}. \tag{9}$$

Based on this lemma, we will construct a proof of the inequality between the Data Scale $D$ and Real Entropy $S'^{real}$ below:

**Theorem A.2.**  *Assuming that the user sequence can be modeled as a first-order aperiodic stationary Markov chain (50). If the user sequence $S = \{S_u, u \in U\}$, then the relationship between the sum of minimum encoding length $|U|C_{min}$ and Real Entropy $S'^{real}$ is given by:*

$$D \sim D' = |U|C_{min} \geq (\Sigma_u |S_u|) \cdot S'^{real}. \tag{10}$$

*Proof.*  From Lemma A.1, it follows that:

$$
\begin{aligned}
|U|C_{min} = \sum_{\forall \Sigma_u S_u} p(\Sigma_u S_u) l_i &= \sum_{\forall \Sigma_u S_u} p(\Sigma_u S_u) log_2 \frac{1}{2^{-l_i}} \\
&\geq H_2(\Sigma_u S_u) = \sum_{\forall \Sigma_u S_u} p(\Sigma_u S_u) \log_2 \frac{1}{p(\Sigma_u S_u)},
\end{aligned} \tag{11}
$$

where $H_2(S_u)$ represents the entropy of the sequence in log-base 2, and Kraft's inequality (68) $\sum_{i=1}^{n} 2^{-l_i} \leq 1$, $l_i$ is encoding length was utilized. Meanwhile, from Lemma 3.1 with Real Entropy $S'^{real} = 1/S^{real}$, it follows that:

$$\frac{\Sigma_u |S_u|}{S^{real}} = \frac{\Sigma_u |S_u|}{\left(\frac{1}{\Sigma_u |S_u|} \sum_j \Lambda_j\right)^{-1} \ln \Sigma_u |S_u|}, \tag{12}$$

Combining Eq.(11) and Eq.(12), the Eq.(10) is equivalent to proving:

$$\sum_{\forall \Sigma_u S_u} p(\Sigma_u S_u) \log_2 \frac{1}{p(\Sigma_u S_u)} \geq \frac{\Sigma_u |S_u|}{\left(\frac{1}{\Sigma_u |S_u|} \sum_j \Lambda_j\right)^{-1} \ln \Sigma_u |S_u|}, \tag{13}$$

where it is necessary to ensure that interaction distribution $p(\Sigma_u S_u)$ is minimal, which is more easily satisfied in recommendations with a large recommendation item vocabulary. Combining the above equations, we need to demonstrate that:

$$\sum_{\forall \Sigma_u S_u} p(\Sigma_u S_u) \log_2 \frac{1}{p(\Sigma_u S_u)} \left(\frac{1}{\Sigma_u |S_u|} \Sigma_j \Lambda_j\right)^{-1} \ln \Sigma_u |S_u| \geq \Sigma_u |S_u|. \tag{14}$$

We decompose inequality (14) into the following **two inequalities (15) and (16)** for our final proof:

$$\left(\frac{1}{\Sigma_u |S_u|} \sum_j \Lambda_j\right)^{-1} \ln \Sigma_u |S_u| \geq \frac{4|U| \ln \Sigma_u |S_u|}{(\Sigma_u |S_u| + 2)}, \tag{15}$$

$$\sum_{\forall \Sigma_u S_u} p(\Sigma_u S_u) \log_2 \frac{1}{p(\Sigma_u S_u)} \geq \frac{\Sigma_u |S_u|(\Sigma_u |S_u| + 2)}{4|U| \ln \Sigma_u |S_u|}. \tag{16}$$

**Proof of Inequality (15)** For the first inequality derived, since j is the subsequence starting from j after concatenating all sequences, the minimum value of $S'^{real}$ is achieved when all sequences in the interaction are identical. In this case, $a$ is a double arithmetic sequence increasing from 1 to $|S_u|/2$ and then returning to 1 in each user |U|. It is formally expressed as follows:

$$\left(\frac{1}{\Sigma_u |S_u|} \sum_j \Lambda_j\right)^{-1} \ln \Sigma_u |S_u| \geq \left(\frac{2}{|U| \Sigma_u |S_u|} \sum_{j/2} \left(\frac{\Sigma_u |S_u|}{2} - j\right)\right)^{-1} \ln \Sigma_u |S_u|$$
$$= \left(\frac{\Sigma_u |S_u|(\Sigma_u |S_u| + 2)}{4|U| \Sigma_u |S_u|}\right)^{-1} \ln \Sigma_u |S_u| = \frac{4|U| \ln \Sigma_u |S_u|}{(\Sigma_u |S_u| + 2)}. \tag{17}$$

Here, it is assumed that $\frac{\ln(\Sigma_u |S_u|)}{\Sigma_u |S_u| + 2} > \frac{1}{|U|}$ when $\forall u, |S_u| \geq 4$, which is reasonable in the context of recommendation systems.

**Proof of Inequality (16)** On the other hand, as the generation probability distribution of most sequences tends to be relatively uniform and considering all permutations of all user sequences, we have:

$$\sum_{\forall \Sigma_u S_u} p(\Sigma_u S_u) \log_2 \frac{1}{p(\Sigma_u S_u)} \sim \frac{1}{p(\Sigma_u S_u)} p(\Sigma_u S_u) \log_2 \frac{1}{p(\Sigma_u S_u)} \sim \frac{\Sigma_u |S_u|}{\ln 2} \ln |I| \tag{18}$$

Due to the differing distributions across datasets, deriving conclusions without considering actual circumstances can be challenging. To address this, we introduce an empirical inequality: $5 \ln |I| \ln \Sigma_u |S_u| \geq |S_u|_{\max} + 2$. The left side of this inequality increases as the dataset grows larger, whereas the right side remains a relatively small constant in typical recommendation tasks (maximum of 500 in our datasets) (69). In our smallest dataset, Kuairand, the left side evaluates to 581, still satisfying the inequality. Therefore, we can assert that this inequality will likely hold under the Scaling Data condition. Under this inequality, we have:

$$\sum_{\forall \Sigma_u S_u} p(\Sigma_u S_u) \log_2 \frac{1}{p(\Sigma_u S_u)} \sim \frac{\Sigma_u |S_u|}{\ln 2} \ln |I| = \frac{5|U| \Sigma_u |S_u| \ln |I| \ln \Sigma_u |S_u|}{5 \ln 2 |U| \ln \Sigma_u |S_u|}$$
$$\geq \frac{(|U|(|S_u|_{\max} + 2)) \Sigma_u |S_u|}{5 \ln 2 |U| \ln \Sigma_u |S_u|} \geq \frac{(\Sigma_u |S_u| + 2) \Sigma_u |S_u|}{5 \ln 2 |U| \ln \Sigma_u |S_u|} \geq \frac{\Sigma_u |S_u|(\Sigma_u |S_u| + 2)}{4|U| \ln \Sigma_u |S_u|} \tag{19}$$

Overall, Inequality (17) proves Inequality (15), and Inequality (19) proves Inequality (16). Thus, we have completed the proof of Inequality (14) and, ultimately, the proof of the theorem. □

## A.2 Proof of Theorem 4.2

After obtaining the model's loss and performance under different data and model parameters, we need to construct a fitting function for the final model performance. However, nearly all sequential recommendation metrics (e.g., NDCG, HR, etc.) are discrete, making it difficult to establish their relationship with the model directly. Therefore, We first select an appropriate metric by examining the differences between performance and testing loss, choosing one with minimal discrepancy. We then analyze the loss constraint inequality during testing. Due to the guarantee of consistency, this inequality can be directly applied to Performance. Ultimately, we introduce the Squeeze Theorem of Performance Fitting to establish a rigorous foundation for understanding the relationship between model configurations and performance. According to the definition of the Squeeze Theorem, we will identify a reasonable fitting function for model performance evaluation. The whole process is shown in Part (C) in Figure 2. Since our analysis focuses on testing losses, our analytical approach still addresses the overfitting challenges faced by the original Scaling Law.

### A.2.1 Metric-Loss Consistency

We then need to discuss the consistency between testing loss and Performance. We chose NDCG and HR as metrics in our experimental analysis to measure the model's performance. Their consistency is ensured by (70), respectively, and possesses the following properties:

$$|L_{\text{NDCG}} - L_{test}| \leq \frac{\Delta_F}{\sqrt{\Delta_\phi}} \cdot \sqrt{\Phi(L_{NDCG}) - \Phi(L_{test})}. \tag{20}$$

Here, $L_{\text{NDCG}}$ represents an NDCG-like function, which is formalized as $-\frac{1}{M} \sum_{j=1}^m \frac{G(r_j)}{F(\omega(j))}$. By replacing weight distribution $\omega(j)$ with the mean value, it becomes HR. $\Phi$ denotes a convex function, while $\Delta_F$ and $\Delta_\phi$ are parameters. The upper bound of this difference ensures the appropriateness of our selected metrics, ensuring that in our subsequent analysis of model performance, the formulas for performance and testing loss allow for interchangeable use.

### A.2.2 Inequaly on Performance Decay

Subsequently, the testing loss, which can also be interpreted as performance due to consistency, is constrained as follows:

**Lemma A.3.** *(48) Suppose input sequence of each user* $\mathbf{e}_u = [\mathbf{e}^{i_1}, \mathbf{e}^{i_2}, \dots, \mathbf{e}^{i_{|S_u|}}] \in R^{d_{In}}$, *then with a small offset $\delta$ we have*

$$1 < L_{test}(H, d_{emb}, D) = \log Z_t + \frac{1}{Z_t} + \log H + \frac{1}{H} - \delta, \tag{21}$$

$$\frac{D \cdot \Psi_{d_{In}}(\sqrt{\frac{d_{In}}{2\pi e}})}{\exp(\sqrt{\frac{d_{In}}{2\pi e}})} \leq Z_t \leq D \cdot \Psi_{d_{In}}(\sqrt{\frac{d_{In}}{2\pi e}}), \tag{22}$$

*where $L_{test}$ denotes testing loss, $\Psi_{d_{In}}(r) = \pi^{\frac{d_{In}}{2}} r^{d_{In}} / \Gamma(1 + \frac{d_{In}}{2})$, $d_{In} = |S_u|_{max} \times d_{emb}$ is the dimension of input sequence.*

we can utilize constraints provided by Lemma A.3 and Eq.(20), which can be formulated as:

$$1 < P(H, d_{emb}, D) = \log Z_t + \frac{1}{Z_t} + \log H + \frac{1}{H} - \delta, \tag{23}$$

where $\delta$ is a small offset including $\frac{\Delta_F}{\sqrt{\Delta_\phi}} \cdot \sqrt{\Phi(L_{NDCG}) - \Phi(L_{test})}$. Next, we will introduce the Squeeze Theorem of Performance Fitting. Overall, we first introduce exponential parameters $w_3$ and $w_4$ for the model depth $H$ and the embedding dimension $d_{emb}$, respectively. We then use the properties of the Squeeze Theorem to prove the existence of these parameters $w_3$ and $w_4$ between lower bounds $\hat{w}_3$ and $\hat{w}_4$, and upper bounds $w_3'$ and $w_4'$. Specifically, we have:

Table 3: The basic information for different datasets, where $|S_u|$ denotes the sequence length, along with the fitted data parameter $D$ in different metrics (HR, NDCG, Loss). It's relationship versus $1/\#Tokens \cdot (S'^{Real})$ and $1/\#Tokens$ is illustrated in Figure 3. $D$ becomes excessively large in Industrial affecting the fitting process, we uniformly take the reciprocal, $1/\cdot$, for the analysis.

| Dataset Details, $\#Tokens = \Sigma_u|S_u|$ | | | | | Dataset Parameter Fitting(*1E-07) | | |
| Dataset | | $|S_u|_{max}$ | $\#Tokens$ | $S'^{Real}$ | $1/D_{HR}$ | $1/D_{NDCG}$ | $1/D_{Loss}$ |
|---|---|---|---|---|---|---|---|
| KR-Pure | User: 27,285 Item: 7,551 | 25 | 447,407 | 0.2998 | 0.9800 | 1.0531 | 0.0352 |
| | | 50 | 570,537 | 0.3153 | 0.8081 | 0.7080 | 0.0235 |
| | | 100 | 661,028 | 0.3544 | 0.5560 | 0.5669 | 0.0199 |
| ML-1M | User: 6,040 Item: 3,706 | 100 | 505,108 | 0.1864 | 1.3699 | 1.5003 | 0.0459 |
| | | 150 | 802,493 | 0.1856 | 0.8972 | 0.8130 | 0.0263 |
| | | 200 | 1,058,511 | 0.1854 | 0.7839 | 0.7039 | 0.0218 |
| AMZ-Books | User: 694,897 Item: 686,623 | 50 | 8,044,865 | 0.1130 | 0.2100 | 0.1693 | 0.0021 |
| | | 25 | 7,076,238 | 0.1129 | 0.2404 | 0.2093 | 0.0030 |
| Industrial | User: 19,252,028 Item: 234,488 | 50 | 513,878,761 | 0.3769 | 0.0142 | 0.0175 | 0.0001 |
| | | 25 | 327,509,107 | 0.4001 | 0.0197 | 0.0249 | 0.0001 |

**Theorem A.4.** *Squeeze Theorem of Performance Fitting.* *There exist* $\hat{w}_3, w'_3, \hat{w}_4, w'_4$ *such that*

$$
log(d_{emb}^{\hat{w}_3}D^{\hat{w}_4}) + \frac{1}{d_{emb}^{\hat{w}_3}D^{\hat{w}_4}} + \log H + \frac{1}{H} - \delta \leq P(H, d_{emb}, D)
$$
$$
\leq log(d_{emb}^{w'_3}D^{w'_4}) + \frac{1}{d_{emb}^{w'_3}D^{w'_4}} + \log H + \frac{1}{H} + \delta. \tag{24}
$$

*Proof.* The lower bound of A.4 is the utilize of the property of function $\frac{1}{\cdot} + log(\cdot)$, which is formulated as:
$$
log(D^0 d_{emb}^0) + \frac{1}{d_{emb}^0 D^0} = 1 \leq log(Z_t) + \frac{1}{Z_t}. \tag{25}
$$

On the other side, the upper bound can be proven using the following inequalities:

$$
Z_t \leq D \cdot \Psi_n(\sqrt{\frac{d_{In}}{2\pi e}}) = \frac{D\pi^{\frac{d_{In}}{2}}\left(\sqrt{\frac{d_{In}}{2\pi e}}\right)^{d_{In}}}{\Gamma(\frac{d_{In}}{2}+1)} \sim k\frac{D}{\sqrt{\frac{S_{max}d_{emb}}{2}\pi e}}
$$
$$
log(Z_t) + \frac{1}{Z_t} \leq O(log(D \cdot d_{emb}^{-\frac{1}{2}}) + \frac{1}{D \cdot d_{emb}^{-\frac{1}{2}}}). \tag{26}
$$

Here, we apply Stirling's approximation $\Gamma(z+1) \sim \sqrt{2\pi z}\left(\frac{z}{e}\right)^z$ as the proof of the rightmost inequality. This requires $Z_t \leq 1$, which is common in the overfitting phenomenon when the model studied in this paper increases rapidly, as evidenced by the u-shape images in the experiments. This approach enables us to decompose the loss into the form $\frac{1}{\cdot} + log(\cdot)$, where $\frac{1}{\cdot}$ and $log(\cdot)$ serve as mutual decay terms, optimizing the fit for performance. Similar to (7; 71), We factorize the product $d_{emb}^{w_3}D^{w_4}$ into $d_{emb}^{w_3}$ and $D^{w_4}$ adding extra parameter $w_1$, $w_2$ and $w_5$ as the final structure of Eq.( 8), which represents the ultimate form of our performance fitting model. $\square$

### A.3 Details on Dataset and Data Parameter Fitting

We present the specific details of the dataset and the precise numerical values for the images shown in Section 5.1.1 in Table 3.

### A.4 Detailed Experiment Settings

Following previous works (55; 56; 57), we leverage the leave-one-out method to calculate the recommendation performance. Besides, we adopt the whole item set as the candidate item set during evaluation to avoid the sampling bias of the candidate selection (72). Then, we evaluate the Top-K recommendation performance by Normalized Discounted Cumulative Gain (NDCG) (59) and Hit

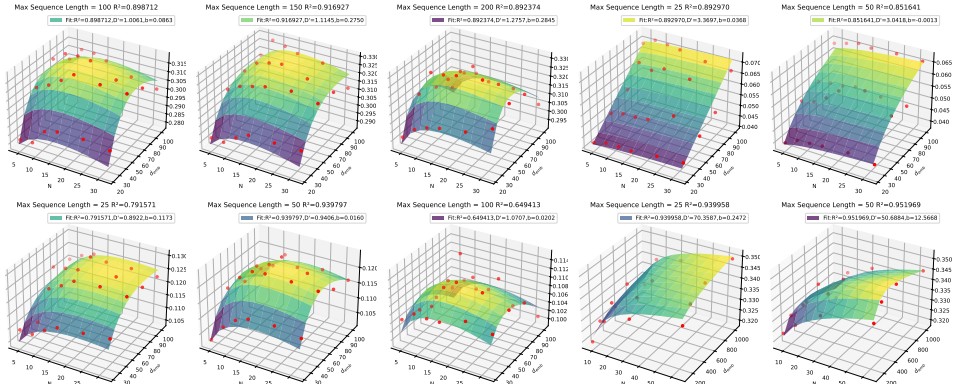

Figure 5: The PL function fitting between model performance (Z-axis, HR) and the number of layers (x-axis, $H$), as well as the embedding dimensions (y-axis, $d_{emb}$), the plot includes annotations of the correlation coefficients $R^2$.

Rate (HR) (58). To effectively demonstrate the performance of models with varying parameters across different datasets, we selected different parameters for fitting based on the size of each dataset. Regarding model configurations, for the MovieLens-1M, KuaiRand-pure, and Amazon Books datasets, we configured $N \in \{4, 8, 12, 16, 24, 32\}$ and $d_{emb} \in \{25, 50, 75, 100\}$. For the Industrial dataset, we set $N \in \{8, 16, 32, 64\}$ and $d_{emb} \in \{128, 256, 512, 1024\}$. From a data perspective, we selected the maximum sequence length for truncation based on the average length of each dataset. In the MovieLens-1M dataset, we selected according to the maximum sequence length $S_{max} \in \{100, 150, 200\}$. In the KuaiRand-Pure dataset, we set the maximum sequence length $S_{max} \in \{25, 50, 100\}$. Finally, for the Amazon Books and Industrial datasets, we configured the maximum sequence length $S_{max} \in \{25, 50\}$. The largest model we executed reached a model depth of $H = 64$, an embedding dimension of $d_{emb} = 1024$, and a vocabulary size of $|I| = 19,252,028$. We utilized 48 industrial GPUs to run this experiment, with the largest experiment taking 24 hours. This truly allowed us to study model performance at extreme data and model scales.

## A.5 Application 2: Exploring Performance Law Potential Among Framework

Another application of the Performance Law is to assess potential performance gains when scaling up the model. We conducted experiments and fitting analyses on three different frameworks (HSTU (10), LLaMA2 (8), and SASRec (15)) evaluated at different precisions: float32 and bfloat16. Experiments were performed on the smaller dataset (ML-1M) to enable the models to more easily reach their optimal upper bounds. In the expressions $w_1(log(H^{w_3}) + \frac{p1}{H^{w_3}})$ and $w_2(log(d_{emb}^{w_4}) +$

Table 4: Comparison of Model Parameters and Performance Across Different Precisions in Movielens-1M with NG denotes NDCG. All results are statistically significant with p<0.05.

| Precision | Float32 | | | Bfloat16 | | |
|---|---|---|---|---|---|---|
| Model | HSTU | LLaMA2 | SASRec | HSTU | LLaMA2 | SASRec |
| $w_1 \uparrow$ | 0.009 | **0.036** | 0.007 | 0.003 | 0.015 | -0.014 |
| $w_2 \uparrow$ | 0.083 | **0.159** | 0.001 | 0.034 | 0.086 | 0.008 |
| HR@10↑ | 0.332 | **0.346** | 0.302 | 0.332 | 0.336 | 0.293 |
| HR@50↑ | 0.585 | **0.598** | 0.573 | 0.594 | 0.598 | 0.561 |
| NG@10↑ | 0.185 | **0.194** | 0.172 | 0.187 | 0.188 | 0.162 |
| NG@50↑ | 0.242 | **0.252** | 0.231 | 0.247 | 0.249 | 0.221 |

$\frac{p1}{d_{emb}^{w_4}}$) in Eq.(8), the upper bound of $log(H^w) + \frac{p1}{H^w}$ does not change with variations in $w$ when $p1$ is fixed. Therefore, larger values of $w_1$ and $w_2$ indicate a better scaling-up potential for the model. We tested several different types of models (HSTU, LLaMA, and SASRec) and different precisions (Float32 and Bfloat16) to observe the relationship between optimal model performance and the fitted parameters $w_1$ and $w_2$. The results are presented in Table 4. From the table, we conclude that the model's performance closely aligns with the magnitude trend of $w_1$ and $w_2$, further underscoring the accuracy of our quantitative fitting. This also demonstrates that the Performance Law can effectively guide model structure configuration, thereby reducing memory and time when modifying frameworks.

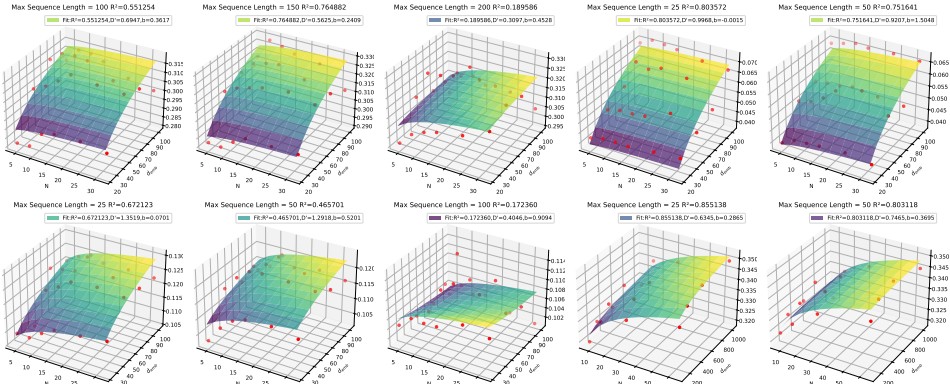

Figure 6: The SL function fitting between model performance (Z-axis, HR) and the number of layers (x-axis, $H$), as well as the embedding dimensions (y-axis, $d_{emb}$), the plot includes annotations of the correlation coefficients $R^2$.

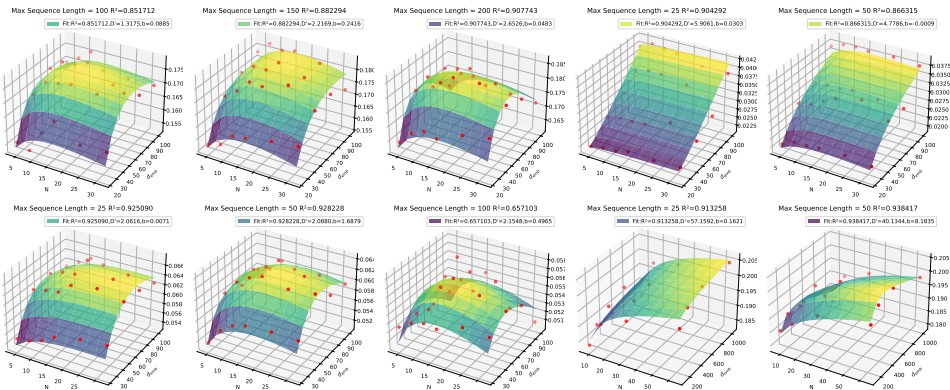

Figure 7: The PL function fitting between model performance (Z-axis, NDCG) and the number of layers (x-axis, $H$), as well as the embedding dimensions (y-axis, $d_{emb}$), the plot includes annotations of the correlation coefficients $R^2$.

### A.6 Result Illusion Extension

## B Additional Model Extension Results

Detailed quantitative results for *Performance Law*, Scaling Law, and Precision Scaling Law on additional advanced models and the Amazon Beauty long-tail scenario are presented below.

As shown, the Performance Law consistently surpasses conventional Scaling Law and Precision Scaling Law in terms of both fitting quality ($R^2$) and error measures (MAE, RMSE), demonstrating its extensibility and robustness across graph-based, sequence-based, multimodal, diffusion-based, and long-tail recommendation models.

## C Experimental Results for Data Quality Evaluation Functions

We provide a detailed comparison of different data quality evaluation functions: Real Entropy, Token Only, Approximate Entropy (64), Shannon Entropy, and Kolmogorov Complexity. The results across three metrics (NG, HR, L) are presented in Table 6, where higher values indicate better fitting ability.

As shown above, Real Entropy achieves the best overall performance as a data quality metric, outperforming Approximate Entropy (64), Shannon Entropy, and Kolmogorov Complexity.

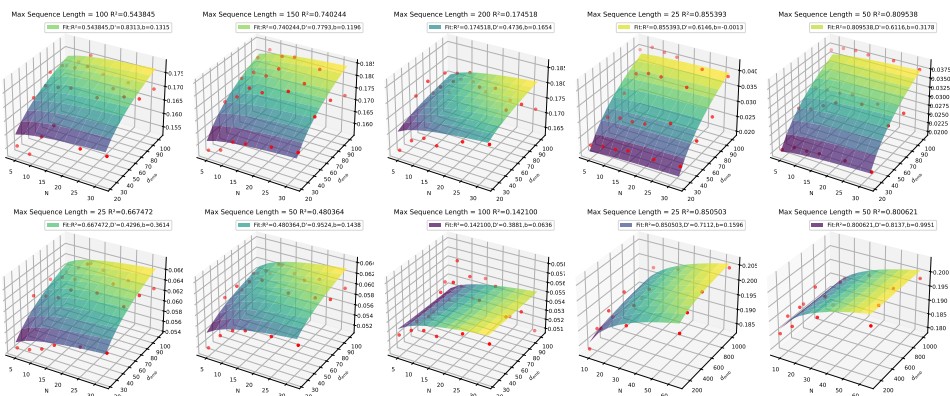

Figure 8: The PL function fitting between model performance (Z-axis, NDCG) and the number of layers (x-axis, $H$), as well as the embedding dimensions (y-axis, $d_{emb}$), the plot includes annotations of the correlation coefficients $R^2$.

## D Additional Evaluation Metrics: MAE and RMSE

**W5:** *The evaluation metric is overly dependent on $R^2$; additional metrics such as MAE/RMSE or downstream utility of predicted parameters would strengthen the empirical support.*

**Answer:** Thank you for your valuable suggestion. To address this concern, we have supplemented the empirical analysis with corresponding MAE and RMSE results for each dataset and model size, in addition to $R^2$. Tables 7 and 8 below report the detailed MAE and RMSE comparisons. The results demonstrate that *Performance Law* consistently achieves lower MAE and RMSE values than both Scaling Law and Precision Scaling Law across all scenarios, supporting its robust predictive accuracy beyond the $R^2$ metric.

Table 5: Performance comparison on LightGCN, Mamba, Wukong, DiffuRec, and Amazon Beauty (long-tail).

| Model | Metric | $R^2 \uparrow$ | MAE (x1e-3)$\downarrow$ | RMSE (x1e-3)$\downarrow$ |
|---|---|---|---|---|
| **LightGCN** | | | | |
| HR@10 | Performance Law | **0.879** | **4.665** | **5.806** |
| | Scaling Law | 0.825 | 5.456 | 6.978 |
| | Precision Scaling Law | 0.866 | 4.918 | 6.116 |
| NDCG@10 | Performance Law | **0.850** | **5.423** | **6.744** |
| | Scaling Law | 0.800 | 6.120 | 7.802 |
| | Precision Scaling Law | 0.806 | 5.797 | 7.223 |
| **Mamba** | | | | |
| HR@10 | Performance Law | **0.818** | **1.941** | **2.512** |
| | Scaling Law | 0.532 | 3.342 | 4.021 |
| | Precision Scaling Law | 0.551 | 3.169 | 3.805 |
| NDCG@10 | Performance Law | **0.810** | **0.398** | **0.530** |
| | Scaling Law | 0.663 | 0.553 | 0.705 |
| | Precision Scaling Law | 0.663 | 0.532 | 0.701 |
| **Wukong** | | | | |
| AUC | Performance Law | **0.528** | **0.624** | **0.868** |
| | Scaling Law | 0.479 | 0.654 | 0.912 |
| | Precision Scaling Law | 0.503 | 0.636 | 0.891 |
| **DiffuRec** | | | | |
| HR@10 | Performance Law | **0.766** | **34.933** | **46.596** |
| | Scaling Law | 0.608 | 50.277 | 60.358 |
| | Precision Scaling Law | 0.652 | 46.882 | 56.908 |
| NDCG@10 | Performance Law | **0.748** | **17.077** | **20.707** |
| | Scaling Law | 0.617 | 21.006 | 25.505 |
| | Precision Scaling Law | 0.666 | 19.339 | 23.818 |
| **Amazon Beauty (Long-tail Rec.)** | | | | |
| HR@10 | Performance Law | **0.868** | **1.857** | **2.271** |
| | Scaling Law | 0.335 | 4.265 | 5.092 |
| | Precision Scaling Law | 0.334 | 4.267 | 5.094 |
| NDCG@10 | Performance Law | **0.890** | **1.329** | **1.668** |
| | Scaling Law | 0.513 | 2.956 | 3.509 |
| | Precision Scaling Law | 0.513 | 2.957 | 3.510 |

Table 6: Comparison of data quality evaluation functions ($R^2$).

| Metric | Real Entropy | Token Only | Approx. Entropy | Shannon Entropy | Kolmogorov Complexity |
|---|---|---|---|---|---|
| NDCG | **0.9906** | 0.8160 | 0.9825 | 0.7982 | 0.8955 |
| HR | **0.9913** | 0.8259 | 0.9774 | 0.7998 | 0.8663 |
| Loss | **0.9881** | 0.8776 | 0.9471 | 0.8617 | 0.9374 |

Table 7: MAE comparison for HR@10 and NDCG@10 across datasets and model sizes.

| Dataset | $|S_u|$ | HR@10 (MAE $\times 10^{-3}$) | | | NDCG@10 (MAE $\times 10^{-3}$) | | |
|---|---|---|---|---|---|---|---|
| | | Perf. Law | Scaling Law | Precision Law | Perf. Law | Scaling Law | Precision Law |
| KR-Pure | 25 | **2.111** | 2.793 | 2.800 | **0.768** | 1.655 | 1.658 |
| | 50 | **1.027** | 3.269 | 3.273 | **0.619** | 1.921 | 1.922 |
| | 100 | **1.871** | 2.587 | 2.588 | **0.958** | 1.458 | 1.459 |
| ML-1M | 100 | **2.990** | 6.064 | 6.086 | **1.928** | 3.927 | 3.934 |
| | 150 | **2.013** | 2.716 | 2.725 | **2.013** | 2.716 | 2.725 |
| | 200 | **1.645** | 4.844 | 4.846 | **1.645** | 4.844 | 4.846 |
| AMZ-Books | 25 | **2.704** | 3.495 | 3.462 | **1.633** | 2.271 | 2.252 |
| | 50 | **2.719** | 2.725 | 2.720 | **1.662** | 1.715 | 1.712 |
| Industrial | 25 | **1.112** | 2.291 | 2.302 | **1.031** | 1.864 | 1.870 |
| | 50 | **0.972** | 3.356 | 3.356 | **0.830** | 2.700 | 2.700 |

Table 8: RMSE comparison for HR@10 and NDCG@10 across datasets and model sizes.

| Dataset | $|S_u|$ | HR@10 (RMSE $\times 10^{-3}$) | | | NDCG@10 (RMSE $\times 10^{-3}$) | | |
|---|---|---|---|---|---|---|---|
| | | Perf. Law | Scaling Law | Precision Law | Perf. Law | Scaling Law | Precision Law |
| KR-Pure | 25 | **2.880** | 3.610 | 3.615 | **0.923** | 1.943 | 1.946 |
| | 50 | **1.260** | 3.754 | 3.758 | **0.806** | 2.170 | 2.171 |
| | 100 | **2.534** | 3.302 | 3.303 | **1.147** | 1.827 | 1.827 |
| ML-1M | 100 | **3.417** | 7.332 | 7.351 | **2.525** | 4.425 | 4.431 |
| | 150 | **2.657** | 3.948 | 3.957 | **2.657** | 3.948 | 3.957 |
| | 200 | **1.912** | 5.493 | 5.494 | **1.912** | 5.493 | 5.494 |
| AMZ-Books | 25 | **3.535** | 4.084 | 4.058 | **2.148** | 2.626 | 2.612 |
| | 50 | **3.527** | 3.610 | 3.609 | **2.117** | 2.173 | 2.171 |
| Industrial | 25 | **1.294** | 2.676 | 2.693 | **1.186** | 2.223 | 2.232 |
| | 50 | **1.341** | 3.857 | 3.865 | **1.140** | 3.108 | 3.111 |

