# OpenReview forum: "P-Law: Predicting Quantitative Scaling Law with Entropy Guidance in Large Recommendation Models"
_NeurIPS.cc/2025/Conference — NeurIPS 2025 poster_

### Official Review · Reviewer_UaaA · 2025-06-19

**Clarity:** 2
**Significance:** 2
**Originality:** 2
**Rating:** 4
**Confidence:** 4

**Summary:**

The paper introduces Performance Law (PL), a novel theoretical framework designed to address the performance prediction challenge in Sequential Recommendation Systems. PL integrates real entropy-based data quality measurement and decay-aware performance modeling to improve the optimization of sequential recommendation models. By leveraging LZ compression for entropy estimation and mathematical modeling of performance decay, the framework provides a principled approach for predicting model performance under different parameter configurations while avoiding the limitations of traditional scaling laws that rely solely on token counts or loss reduction. The proposed method demonstrates particular effectiveness in small-data scenarios and industrial-scale applications through rigorous experimentation.

**Questions:**

1. The experiments only include a comparison between SL and PcSL methods. It remains unclear why recommendation algorithms based on SL were not included in the evaluation. Could the authors clarify the distinctions between the proposed PL approach and other SL-based recommendation algorithms, such as Wukong? A detailed comparison would help elucidate the unique contributions and advantages of PL.
2. The paper demonstrates the effectiveness of Real Entropy for filtering redundant sequences, but does not compare it with other information-theoretic measures (e.g., Shannon entropy or Kolmogorov complexity) that could similarly quantify sequence diversity. Why were these alternatives not explored? Additionally, how does the choice of LZ compression—instead of other complexity estimators—affect the robustness of Real Entropy in noisy or non-stationary user behavior scenarios?
3. While the Performance Law shows empirical results on datasets like MovieLens and Industrial, its reliance on Real Entropy calculation via LZ compression may introduce computational bottlenecks. For real-time industrial systems processing billions of sequences, what optimizations (e.g., approximate algorithms or parallelization) are proposed to ensure scalability? Furthermore, how would the method handle dynamically evolving user interactions where Real Entropy needs incremental updates?

**Ethical Concerns:**

["NO or VERY MINOR ethics concerns only"]

**Final Justification:**

Since most of the concerns have been discussed, I will improve the rating to 4 in the final output. For example, authors provide more experiments to respond to our question, which is a good sign.

**Limitations:**

YES

**Quality:**

2

**Strengths And Weaknesses:**

1. This method introduces Real Entropy to measure the information content of user behavior sequences, addressing the issue of redundancy.
2. Unlike traditional scaling laws that rely on fitting loss values, this approach fits performance metrics of recommendation systems, which is more applicable and rational in industrial scenarios.
3. Experimental results demonstrate performance improvements, particularly on small datasets.

Weaknesses:
1. Although an approximate LZ compression algorithm is proposed, the computational cost may still be high in large-scale online settings, which is unfavorable for industrial applications.
2. The proposed method involves a substantial number of hyperparameters, including those related to entropy estimation, model architecture, and performance metric fitting. The sensitivity of the method to these hyperparameters can pose challenges in practical deployment. Moreover, the optimal configuration is likely to vary significantly across different domains, datasets, and system constraints, necessitating extensive tuning efforts that may hinder scalability and generalizability.
3. The experimental evaluation in this study lacks testing in several important special scenarios, such as graph-based recommendation and long-tail user segments. These scenarios often present unique challenges and are highly relevant in real-world applications. The absence of such evaluations makes it difficult to ascertain the effectiveness and generalizability of the proposed method in these complex or less typical settings.
4. The experiments only compare the proposed method with general scaling law approaches, without comparing it to recommendation algorithms based on scaling laws. As such, the superiority of this method in recommendation systems remains unsubstantiated.

---

> ### Author Rebuttal · Authors · 2025-07-31
>
> We sincerely thank you for your valuable comments. We hope the following responses can address your concerns.
>
> >  W1 Although an approximate LZ compression algorithm is proposed, the computational cost may still be high in large-scale online settings, which is unfavorable for industrial applications
>
>  We thank the reviewer for pointing out the effectiveness trade-off. We have extended the fragmented averaging strategy to accelerate computations on industrial datasets by segmenting the original dataset into fragments of 100,000 sequences each, and then calculating their averages. This slicing strategy allows for full parallelization.
>
> For the industrial dataset we used, which contains 513,878,761 sequences, the computation time was 1 hour and 32 minutes. This is relatively insignificant compared to the training time of recommendation models of a similar scale. For the 7,076,238-sequence Amazon Books dataset, the computation time was 15 minutes. (Due to parallel computation, the time does not scale strictly proportionally with the size.)
>
> Additionally, this method is advantageous for updates, as new sequences can be simply trimmed to the same fragment size. To demonstrate the stability of fragmented averaging, we present the relationship between the computed values and fragment sizes as follows:
>
> | Segment Length | 100    | 1000   | 10000  | 100000 | All    |
> | -------------- | ------ | ------ | ------ | ------ | ------ |
> | Real Entropy   | 0.2173 | 0.1464 | 0.1142 | 0.1132 | 0.1130 |
>
>  It can be observed that when the slice count exceeds 10,000, the results are already quite close to the true value.  This demonstrates the rationality of using fragmented approximation.
>
> >  W2.1 The sensitivity of the method to these hyperparameters can pose challenges in practical deployment.
>
> Thank you for your attention to parameter sensitivity. For each model parameter (such as the number of layers \( N \) and data \( D \)), our approach introduces only one additional hyperparameter compared to the Scaling Law: \( p1 \) for the model and \( p2 \) for the data. We fixed these additional parameters at 1 to align with the sensitivity of the Scaling Law and conducted the following comparison: the fitting capability differs from the original Performance Law by at most 0.5%, still outperforming other fitting approaches.
>
> | Dataset    | $\|S_u\|$ | Performance Law($R^2$) | Scaling Law($R^2$) | Precision Scaling Law($R^2$) |
> | ---------- | --------- | ---- | :-- | - |
> | KR-Pure    | 25        | **0.791**    | 0.672    | 0.672  |
> |            | 50        | **0.938**      | 0.466     | 0.466      |
> |            | 100       | **0.648**     | 0.172        | 0.182  |
> | ML-1M      | 100       | **0.898**     | 0.551              | 0.699 |
> |            | 150       | **0.916**  | 0.765              | 0.757   |
> |            | 200       | **0.890**     | 0.190              | 0.200 |
> | AMZ-Books  | 25        | **0.892**   | 0.803              | 0.837    |
> |            | 50        | **0.851**              | 0.751              | 0.830  |
> | Industrial | 25        | **0.939**              | 0.855              | 0.857  |
> |            | 50        | **0.948**  | 0.803        | 0.767    |
>
> > W2.2 Moreover, the optimal configuration is likely to vary significantly across different domains
>
>  The Scaling Law also varies significantly across different domains; however, it has found widespread application. Our shared aim with the Scaling Law is to predict model performance in each domain using the Performance Law fitted from small-scale experiments. The key difference between the Performance Law and the Scaling Law is that, rather than relying on the loss, the Performance Law directly fits the model's effectiveness.
>
> > W3 The experimental evaluation in this study lacks testing in several important special scenarios, such as graph-based recommendation and long-tail user segments.
>
>  Thank you for your suggestion on validation towards important special scenarios. We have included validation under LightGCN, as well as verification using the Amazon Beauty data, which is widely utilized in enhancing long-tail recommendations \[1\]\[2\]. Performance Law still outperforms other fitting approaches in these scenarios.
>
> \[1\]Liu Q, Wu X, Wang Y, et al. Llm-esr: Large language models enhancement for long-tailed sequential recommendation[J]. Advances in Neural Information Processing Systems, 2024, 37: 26701-26727.
>
> \[2\]Yang, K., Zheng, S., Li, T., Li, X., & Li, H. (2025). GENPLUGIN: A Plug-and-Play Framework for Long-Tail Generative Recommendation with Exposure Bias Mitigation.
>
> | LightGCN                  |                       |$R^2\uparrow$ | MAE (x1e-3)$\downarrow$ | RMSE (x1e-3)$\downarrow$ |
> | ----- | ---- | --- | -- | --- |
> | [HR@10](mailto:HR@10)     | Performance Law       | **0.879**     | **4.665**                | **5.806**                 |
> |  | Scaling Law           | 0.825         | 5.456                    | 6.978                     |
> | | Precision Scaling Law | 0.866         | 4.918                    | 6.116                     |
> | [NDCG@10](mailto:NDCG@10) | Performance Law       | **0.850**     | **5.423**                | **6.744**                 |
> || Scaling Law           | 0.800         | 6.120                    | 7.802                     |
> |  | Precision Scaling Law | 0.806         | 5.797                    | 7.223                     |
>
>
> | long-tail recommendation  |                       | $R^2\uparrow$ | MAE (x1e-3)$\downarrow$ | RMSE (x1e-3)$\downarrow$ |
> | ---- | - | -------- | -- | ----- |
> | [HR@10](mailto:HR@10)     | Performance Law       | **0.868**         | **1.857**   | **2.271**    |
> |   | Scaling Law           | 0.335         | 4.265                    | 5.092                     |
> | | Precision Scaling Law | 0.334         | 4.267                    | 5.094                     |
> | [NDCG@10](mailto:NDCG@10) | Performance Law       | **0.890**         | **1.329**                    | **1.668**   |
> |                           | Scaling Law           | 0.513         | 2.956                    | 3.509                     |
> |                           | Precision Scaling Law | 0.513         | 2.957                    | 3.510                     |
>
>
>
> > W4 The experiments only compare the proposed method with general scaling law approaches, without comparing it to recommendation algorithms based on scaling laws.
>
> Answer: Thank you for your reminder! Please see Q1.
>
>
>
> > Q1 It remains unclear why recommendation algorithms based on SL were not included in the evaluation. Could the authors clarify the distinctions between the proposed PL approach and other SL-based recommendation algorithms, such as Wukong?
>
> The primary focus of Performance Law and Scaling Law is to provide theoretical proofs for scalability, setting them apart from frameworks like Wukong and HSTU, which primarily aim to enhance framework effect and stability.
>
> Our main goal is to improve the predictive capabilities of frameworks under varying parameters. Both Performance Law and Scaling Law can demonstrate their effectiveness on different frameworks. This is evidenced in the Wukong table below, which includes examples such as Wukong and LightGCN in W3, as well as the transformer from our original paper. Notably, on the Wukong framework, Performance Law still outperforms both variations of Scaling Laws.
> | Wukong |                       | $R^2\uparrow$ | MAE (x1e-3)$\downarrow$ | RMSE (x1e-3)$\downarrow$ |
> | ------ | -----| ------------- | ---- | -- |
> | AUC    | Performance Law       | **0.528**         | **0.624**                    | **0.868**                     |
> |        | Scaling Law           | 0.479         | 0.654                    | 0.912                     |
> |        | Precision Scaling Law | 0.503         | 0.636                    | 0.891                     |
>
>
>
> > Q2.1 Why does not compare it with other information-theoretic measures (e.g., Shannon entropy or Kolmogorov complexity)
>
> Apologies for the misunderstanding. This is because Shannon entropy and Kolmogorov complexity are used to compute the entropy of a set, not a sequence. To demonstrate the robustness of our model, we conducted additional experiments replacing Real Entropy with Approximate Entropy [3], Shannon Entropy, and Kolmogorov complexity. Real Entropy maintained optimal fitting capability.
>
> |      | Real Entropy | Token Only | Approximate Entropy | Shannon Entropy | Kolmogorov Complexity |
> | ---- | ------- | ---------- | -- | ---- | ------ |
> | NDCG   | **0.9906**       | 0.8160      | 0.9825  | 0.7982          | 0.8955                |
> | HR   | **0.9913**      | 0.8259     | 0.9774 | 0.7998          | 0.8663                |
> | Loss    | **0.9881**     | 0.8776     | 0.9471   | 0.8617          | 0.9374                |
>
> \[3\]S.M. Pincus, Approximate entropy as a measure of system complexity., Proc. Natl. Acad. Sci. U.S.A. 88 (6) 2297-2301, PNAS 1991.
>
> >Q2.2 Additionally, how does the choice of LZ compression—instead of other complexity estimators—affect the robustness of Real Entropy in noisy or non-stationary user behavior scenarios?
>
> Thank you for pointing out any confusion regarding the concept of approximation. LZ compression is not an approximation; it is equivalent to the original computation. Therefore, it has no impact on robustness, noise, or non-stationary user behavior scenarios. We discussed our extension on its approximation towards computation cost in W1.
>
> > Q3.1 LZ compression may introduce computational bottlenecks.
>
> Our method's computational complexity is entirely adequate for adoption in industrial scenarios, as referenced in W1.
>
> > Q3.2 Furthermore, how would the method handle dynamically evolving user interactions where Real Entropy needs incremental updates?
>
>  As stated at the end of W1, this method is also quite advantageous for updates, as new sequences only need to be trimmed to the same fragment size, or the user's corresponding fragment can be replaced.

---

### Official Review · Reviewer_UySy · 2025-07-03

**Clarity:** 3
**Significance:** 2
**Originality:** 3
**Rating:** 4
**Confidence:** 3

**Summary:**

This paper introduces Performance Law, a novel framework for quantitatively modeling and predicting the performance of sequential recommendation (SR) models. It addresses key limitations of prior Scaling Laws (SLs) in SR by (1) introducing Real Entropy to evaluate data quality and eliminate redundancy, and (2) incorporating a decay term to resolve the loss-performance discrepancy caused by overfitting. Extensive experiments demonstrate superior predictive accuracy over SL baselines. The approach is also shown to be useful for optimal parameter search and model expansion analysis.

**Questions:**

see weakness

**Ethical Concerns:**

["NO or VERY MINOR ethics concerns only"]

**Final Justification:**

I believe the authors have provided a strong rebuttal, effectively addressing my concerns. I think it’s a good paper, so I will maintain my positive evaluation and score.

**Limitations:**

Yes.

**Paper Formatting Concerns:**

Conforms to NeurIPS formatting. Figures and tables are legible.

**Quality:**

2

**Strengths And Weaknesses:**

## Strengths
- Proposes a new *Performance Law* that directly predicts model performance (e.g., HR, NDCG), rather than just training loss, which is more relevant to recommendation quality.
- Introduces Real Entropy, a well-justified and empirically effective data quality measure tailored for sequential recommendation data.
- Adds a decay term to the performance fitting function to model overfitting phenomena, which improves the predictive power.
- Provides strong theoretical analysis with formal definitions, theorems (e.g., Theorem 4.1, 4.2), and derivations.
- Offers extensive experiments on multiple datasets (including a massive industrial one) and consistent gains over SL and PcSL in R² performance.
- Demonstrates practical utility by using the proposed law for optimal parameter search under different constraints and model scaling strategies.
- Paper is well-organized, clearly written, and code is available.

---

### Weaknesses
- The claim that Performance Law is generalizable beyond SR is not empirically supported; all experiments are limited to sequential recommendation tasks.
- The derivation of Theorem 4.1 depends on a first-order stationary Markov assumption for user sequences, which may not hold in real-world applications.
- The baseline comparison is narrow, mainly focusing on SL variants; recent performance estimation techniques are not included.
- The efficiency (runtime, memory) of computing Real Entropy, especially on industrial datasets, is not discussed.
- The evaluation metric is overly dependent on R²; additional metrics such as MAE/RMSE or downstream utility of predicted parameters would strengthen the empirical support.

---

> ### Author Rebuttal · Authors · 2025-07-31
>
> We sincerely thank you for reviewing our manuscript. We hope the following responses can address your concerns.
> > W1 The claim that Performance Law is generalizable beyond SR is not empirically supported; all experiments are limited to sequential recommendation tasks.
>
> Thank you for your advice! We evaluated the predictive performance of the Performance Law in graph learning (LightGCN) and content-based recommendation (Wukong), where it continues to demonstrate leading capabilities.
>
> | LightGCN                  |                       | R^2$\uparrow$ | MAE (x1e-3)$\downarrow$ | RMSE (x1e-3)$\downarrow$ |
> | ------------------------- | --------------------- | ------------- | ----------------------- | ------------------------ |
> | [HR@10](mailto:HR@10)     | Performance Law       | **0.879**     | **4.665**               | **5.806**                |
> |                           | Scaling Law           | 0.825         | 5.456                   | 6.978                    |
> |                           | Precision Scaling Law | 0.866         | 4.918                   | 6.116                    |
> | [NDCG@10](mailto:NDCG@10) | Performance Law       | **0.850**     | **5.423**               | **6.744**                |
> |                           | Scaling Law           | 0.800         | 6.120                   | 7.802                    |
> |                           | Precision Scaling Law | 0.806         | 5.797                   | 7.223                    |
>
> | Wukong |                       | R^2$\uparrow$ | MAE (x1e-3)$\downarrow$ | RMSE (x1e-3)$\downarrow$ |
> | ------ | --------------------- | ------------- | ----------------------- | ------------------------ |
> | AUC    | Performance Law       | **0.528**     | **0.624**               | **0.868**                |
> |        | Scaling Law           | 0.479         | 0.654                   | 0.912                    |
> |        | Precision Scaling Law | 0.503         | 0.636                   | 0.891                    |
>
>
> > W2 First-order stationary Markov assumption for user sequences may not hold in real-world applications.
>
> The Markov assumption for user sequences is considered reasonable in the recommendation domain \[1\]\[2\], making its application in real-world scenarios appropriate as well. We introduce this condition mainly to ensure that the probability of a user's next item interaction can be calculated for all users. Since the Real Entropy for all datasets can be computed, this condition can effectively be overlooked.
>
> [1] Rendle S, Freudenthaler C, Schmidt-Thieme L. Factorizing personalized Markov chains for next-basket recommendation[C]//Proceedings of the 19th international conference on World Wide Web. 2010: 811-820.
> [2] Yang Y, Jang H J, Kim B. A hybrid recommender system for sequential recommendation: combining similarity models with Markov chains[J]. IEEE Access, 2020, 8: 190136-190146.
>
> > W3 The baseline comparison is narrow, mainly focusing on SL variants; recent performance estimation techniques are not included？
>
> Apologize for any misunderstanding. Recent research on large recommendation models has focused on model-based rather than theoretical approaches, with all enhanced framework recommendation models being guided by the Scaling Law. We have already discussed the comparison with the latest Precision Scaling Law [3], as shown in W5.
>
> [3]T. Kumar, Z. Ankner, B. F. Spector, B. Bordelon, N. Muennighoff, M. Paul, C. Pehlevan, C. Ré,494 and A. Raghunathan, “Scaling laws for precision,” arXiv preprint arXiv:2411.04330, 2024
>
> > W4 The efficiency (runtime, memory) of computing Real Entropy, especially on industrial datasets, is not discussed.
>
> We thank the reviewer for pointing out the effectiveness trade-off. We extende our computation on averaging approach to expedite computations on industrial datasets by segmenting the original dataset into shards of 100,000 sequences each and calculating their average values.
>
> This segmentation strategy allows for complete parallelization, resulting in a computation time of 1 hour and 32 minutes for the industrial dataset containing 513,878,761 sequences. This is relatively efficient compared to the training time of recommendation models of this scale. For the Amazon Books dataset, which contains 7,076,238 sequences, the computation takes 15 minutes. (Due to parallel computing, the time is not strictly proportional to the dataset size.)
>
> This method also facilitates updates, as new sequences just need to be divided into segments of the same size. To demonstrate the stability of sharded averaging, we enumerate the relationship between the calculated values and shard size as follows:
>
> | Segment Length | 100    | 1000   | 10000  | 100000 | All    |
> | -------------- | ------ | ------ | ------ | ------ | ------ |
> | Real Entropy   | 0.2173 | 0.1464 | 0.1142 | 0.1132 | 0.1130 |
>
>  It can be observed that when the slice count exceeds 10,000, the results are already quite close to the true value. This demonstrates the rationality of using fragmented approximation.
>
>
>
> > W5 The evaluation metric is overly dependent on R²; additional metrics such as MAE/RMSE or downstream utility of predicted parameters would strengthen the empirical support.
>
>  Thank you for your valuable  suggestion. We have supplemented the paper with all the compared MAE/RMSE values, and we still outperform the remaining Scaling Law analyses.
>
>
>
> |            |      | MAE (HR@10) (x1e-3) |             |                       | MAE (NDCG@10) (x1e-3) |             |                       |
> | ---------- | ---- | -------------------- | ----------- | --------------------- | ---------------------- | ----------- | --------------------- |
> | Dataset    |      | Performance Law      | Scaling Law | Precision Scaling Law | Performance Law        | Scaling Law | Precision Scaling Law |
> | KR-Pure    | 25   | **2.111** | 2.793 | 2.800 | **0.768** | 1.655 | 1.658 |
> |            | 50   | **1.027** | 3.269 | 3.273 | **0.619** | 1.921 | 1.922 |
> |            | 100  | **1.871** | 2.587 | 2.588 | **0.958** | 1.458 | 1.459 |
> | ML-1M      | 100  | **2.990** | 6.064 | 6.086 | **1.928** | 3.927 | 3.934 |
> |            | 150  | **2.013** | 2.716 | 2.725 | **2.013** | 2.716 | 2.725 |
> |            | 200  | **1.645** | 4.844 | 4.846 | **1.645** | 4.844 | 4.846 |
> | AMZ-Books  | 25   | **2.704** | 3.495 | 3.462 | **1.633** | 2.271 | 2.252 |
> |            | 50   | **2.719** | 2.725 | 2.720 | **1.662** | 1.715 | 1.712 |
> | Industrial | 25   | **1.112** | 2.291 | 2.302 | **1.031** | 1.864 | 1.870 |
> |            | 50   | **0.972** | 3.356 | 3.356 | **0.830** | 2.700 | 2.700 |
>
>
>
> |            |      | RMSE (HR@10) (x1e-3) |             |                       | RMSE (NDCG@10) (x1e-3) |             |                       |
> | ---------- | ---- | -------------------- | ----------- | --------------------- | ---------------------- | ----------- | --------------------- |
> | Dataset    |      | Performance Law      | Scaling Law | Precision Scaling Law | Performance Law        | Scaling Law | Precision Scaling Law |
> | KR-Pure    | 25   | **2.880**            | 3.610       | 3.615                 | **0.923**              | 1.943       | 1.946                 |
> |            | 50   | **1.260**            | 3.754       | 3.758                 | **0.806**              | 2.170       | 2.171                 |
> |            | 100  | **2.534**            | 3.302       | 3.303                 | **1.147**              | 1.827       | 1.827                 |
> | ML-1M      | 100  | **3.417**            | 7.332       | 7.351                 | **2.525**              | 4.425       | 4.431                 |
> |            | 150  | **2.657**            | 3.948       | 3.957                 | **2.657**              | 3.948       | 3.957                 |
> |            | 200  | **1.912**            | 5.493       | 5.494                 | **1.912**              | 5.493       | 5.494                 |
> | AMZ-Books  | 25   | **3.535**            | 4.084       | 4.058                 | **2.148**              | 2.626       | 2.612                 |
> |            | 50   | **3.527**            | 3.610       | 3.609                 | **2.117**              | 2.173       | 2.171                 |
> | Industrial | 25   | **1.294**            | 2.676       | 2.693                 | **1.186**              | 2.223       | 2.232                 |
> |            | 50   | **1.341**            | 3.857       | 3.865                 | **1.140**              | 3.108       | 3.111                 |

---

> > ### Comment · Reviewer_UySy · 2025-08-06
> > **Official comments by reviewer UySy**
> >
> > Thank you to the authors for the rebuttal, which addressed my concerns. Therefore, I will keep my score unchanged.

---

### Official Review · Reviewer_7fJJ · 2025-07-03

**Clarity:** 2
**Significance:** 2
**Originality:** 3
**Rating:** 4
**Confidence:** 2

**Summary:**

This paper proposes the Performance Law, highlighting that a reduction in loss does not always lead to improvements in performance. The authors aim to address this issue by introducing Real Entropy as a corrective factor. Compared to existing scaling laws, the proposed approach demonstrates higher prediction accuracy and designs a new objective function capable of explaining overfitting phenomena.

**Questions:**

1. Since the experiments mainly focus on transformer-based models, I am curious whether the proposed formulation would also apply to other types of recommendation models (e.g., RNN, GNN, Mamba).
2. Have you conducted any experiments on recommendation models based on large language models (LLMs)?

**Ethical Concerns:**

["NO or VERY MINOR ethics concerns only"]

**Final Justification:**

The concern about the lack of non-Transformer results was well addressed in the rebuttal. I find the main novelty of this paper to be its focus on the underexplored relationship between model size and performance in recommender systems.

**Limitations:**

Yes

**Paper Formatting Concerns:**

No formatting concerns

**Quality:**

3

**Strengths And Weaknesses:**

- **Quality**: I highly appreciate the motivation to address the limitations of scaling laws, which do not translate well to sequential recommendation tasks.
- **Clarity**: Overall, the paper is clearly written, but the heavy reliance on mathematical formulations makes it challenging to grasp the intuition behind the approach.
- **Significance**: Given the increasing prevalence of large-scale models in recommendation systems, establishing a quantitative framework for performance prediction is likely to have a positive impact on the field.
- **Originality**: I found the topic of quantitatively predicting the performance gains from scaling sequential recommendation models to be novel and interesting.

---

> ### Author Rebuttal · Authors · 2025-07-31
>
> We are grateful for your kind remarks. We hope the following responses can address your remaining concerns.
> > Q1 Since the experiments mainly focus on transformer-based models, I am curious whether the proposed formulation would also apply to other types of recommendation models (e.g., RNN, GNN, Mamba)
>
> Answer: We appreciate your feedback! We have added a comparison of Performance Law within the LightGCN and Mamba frameworks, where it continues to outperform other Scaling Laws.
>
> | LightGCN                  |                       | $R^2\uparrow$ | MAE (x1e-3)$\downarrow$ | RMSE (x10e-3)$\downarrow$ |
> | ------------------------- | --------------------- | ------------- | ----------------------- | ------------------------- |
> | [HR@10](mailto:HR@10)     | Performance Law       | **0.879**     | **4.665**               | **5.806**                 |
> |                           | Scaling Law           | 0.825         | 5.456                   | 6.978                     |
> |                           | Precision Scaling Law | 0.866         | 4.918                   | 6.116                     |
> | [NDCG@10](mailto:NDCG@10) | Performance Law       | **0.850**     | **5.423**               | **6.744**                 |
> |                           | Scaling Law           | 0.800         | 6.120                   | 7.802                     |
> |                           | Precision Scaling Law | 0.806         | 5.797                   | 7.223                     |
>
> | Mamba                     |                       | $R^2\uparrow$ | MAE (x10e-3)$\downarrow$ | RMSE (x10e-3)$\downarrow$ |
> | ------------------------- | --------------------- | ------------- | ------------------------ | ------------------------- |
> | [HR@10](mailto:HR@10)     | Performance Law       | **0.818**     | **1.941**                | **2.512**                 |
> |                           | Scaling Law           | 0.532         | 3.342                    | 4.021                     |
> |                           | Precision Scaling Law | 0.551         | 3.169                    | 3.805                     |
> | [NDCG@10](mailto:NDCG@10) | Performance Law       | **0.810**     | **0.398**                | **0.530**                 |
> |                           | Scaling Law           | 0.663         | 0.553                    | 0.705                     |
> |                           | Precision Scaling Law | 0.663         | 0.532                    | 0.701                     |
>
> >Q2 Have you conducted any experiments on recommendation models based on large language models (LLMs)?
>
> In recommendation systems, LLMs are often utilized through fine-tuning, which differs from the large recommendation models studied in this paper. However, some studies [1] have mentioned that the input sequence length can also affect scaling outcomes. Therefore, we conducted a scaling analysis with sequences of similar user input lengths, as shown in the table below. The performance law still exhibits superior predictive capability compared to the scaling law.
>
> | LLM                       |                       | $R^2\uparrow$ | MAE (x10e-3)$\downarrow$ | RMSE (x10e-3)$\downarrow$ |
> | ------------------------- | --------------------- | ------------- | ------------------------ | ------------------------- |
> | [HR@10](mailto:HR@10)     | Performance Law       | **0.862**     | **0.195**                | **0.253**                 |
> |                           | Scaling Law           | 0.841         | 0.221                    | 0.271                     |
> |                           | Precision Scaling Law | 0.847         | 0.217                    | 0.265                     |
> | [NDCG@10](mailto:NDCG@10) | Performance Law       | **0.875**     | **0.407**                | **0.477**                   |
> |                           | Scaling Law           | 0.555         | 0.844                    | 0.890                     |
> |                           | Precision Scaling Law | 0.555         | 0.849                    | 0.890                     |
>
> \[1\]Yue, Z., Zhuang, H., Bai, A., Hui, K., Jagerman, R., Zeng, H., Qin, Z., Wang, D., Wang, X., & Bendersky, M. (2025). Inference Scaling for Long-Context Retrieval Augmented Generation. ICLR 2025

---

> > ### Comment · Reviewer_7fJJ · 2025-08-01
> >
> > Thank you for the clarification and the extended experiments. I have one follow-up question.
> >
> > I noticed that you provided results across various architectures (e.g., GNN, Mamba, Diffusion), and that the reported R^2 values vary noticeably between them. Could you clarify the cause of these discrepancies? I agree that the proposed performance law shows clear advantages over traditional scaling laws, especially in predicting actual performance. However, since sequential recommendation involves a wide variety of architectures with different inductive biases—unlike language modeling where Transformer-based models dominate—I am curious whether the current formulation, including the decay term, is intended to be architecture-agnostic, or if you foresee the need for architecture-specific adaptations.

---

> > > ### Author Response · Authors · 2025-08-01
> > >
> > > Thank you for your willingness to explore Performance Law further. It is entirely normal for the R^2 values, which measure the degree of correlation, to vary across different architectures. This variation does not imply that Performance Law is an architecture-specific theory. Just as a model might perform differently on various datasets (e.g., achieving an HR@10 of 0.3 on MovieLens and 0.1 on Amazon Books), it is also normal for a theory to exhibit varying R^2 fitting performance across different frameworks. This doesn't mean that a model is only applicable to one specific dataset, nor does it mean that Performance Law applies only to a single kind of architecture.

---

> > > > ### Comment · Reviewer_7fJJ · 2025-08-06
> > > >
> > > > Thanks for the clarification. I'll maintain my positive score.

---

### Official Review · Reviewer_HQgL · 2025-07-03

**Clarity:** 3
**Significance:** 3
**Originality:** 3
**Rating:** 5
**Confidence:** 4

**Summary:**

This paper introduces the Performance Law for sequential recommendation algorithms, aiming to predicts model performance across various settings, intending to provide a quantitative framework for guiding the parameter optimization of future models.
It first utilizes Real Entropy to measure data quality, aiming to remove the low-quality influence of low-entropy redundant sequences. It then investigates a fitting decay term to facilitate the prediction of the major loss-performance discrepancy phenomena of overfitting, ultimately achieving quantitative performance prediction. Extensive experiments on diverse datasets demonstrate quantitative improvements over established scaling law baselines in both performance prediction and parameter optimization tasks.

**Questions:**

1. Can you discuss about the feasibility of the proposed Performance Law in other other SR architectures / advanced generative techniques?
2. It would be beneficial to include additional entropy-based data quality metrics for comparison.

**Ethical Concerns:**

["NO or VERY MINOR ethics concerns only"]

**Final Justification:**

All of my concerns have been addressed by the authors, and they have also released all the code from the rebuttal period in their own anonymous GitHub repository for future researchers to study. Therefore, I have decided to raise my initial score.

**Limitations:**

Please refer to the weakness and questions. btw, I think this work is interesting, I would like to raise my score if the authors can solve my concerns.

**Paper Formatting Concerns:**

NULL

**Quality:**

3

**Strengths And Weaknesses:**

Strengths:
1. The proposed Performance Law is the first approach to move scaling law analysis in SR from qualitative into quantitative domain to anticipate the effects of how both data and model parameters change will guide the optimization of future models.
2. The proposed Performance laws are grounded in theoretical claims with explicit bounds and relationships.
3.  Extensive experiments span several benchmarks and an industrial-scale dataset, demonstrating the effectiveness of Performance Law by displaying exceptional quantitative prediction ability against qualitative SL.
4. The authors have publicly released the dataset and corresponding code.

Weaknesses:
1. The proposed Performance Law focus on scaling Transformer-based models. It is unclear whether the law applies equally well to other SR architectures (e.g., RNN- or GNN-based SR in Sec. 2) / advanced generative techniques (e.g., DM-based SR).
[1] Diffurec: A diffusion model for sequential recommendation
[2] Plug-in diffusion model for sequential recommendation
[3] Generate what you prefer: Reshaping sequential recommendation via guided diffusion
2. The definitions of S^{real} (Real Entropy) and S^{\prime real} (the final measure of Real Entropy) are overly similar, and both are referred to as “Real Entropy” in Lines 213 and 215. The authors could use distinct symbols or notation to clearly differentiate between them.
3. There are only the comparison between #Tokens·S^{\prime real} and #Tokens in In Figure 3. Including additional entropy-based data quality metrics in the comparison could further support the validity of the proposed Performance Law.

---

> ### Author Rebuttal · Authors · 2025-07-31
>
> We sincerely thank you for your valuable comments. We hope the following responses can address your concerns.
> >W1  It is unclear whether the law applies equally well to other SR architectures (e.g., RNN- or GNN-based SR in Sec. 2) / advanced generative techniques (e.g., DM-based SR).
>
> Thanks for the suggestions on validation for different structures. We have provided a comparison of the predictive performances between the LightGCN[1] and DiffuRec[2]. Overall, our results still surpass those of the Scaling Law and Precision Scaling Law.
>
> [1]He, X., Deng, K., Wang, X., Li, Y., Zhang, Y., & Wang, M. (2020). LightGCN: Simplifying and Powering Graph Convolution Network for Recommendation. SIGIR'20
>
> [2]Li Z, Sun A, Li C. Diffurec: A diffusion model for sequential recommendation[J]. ACM Transactions on Information Systems, 2023, 42(3): 1-28.
>
> | LightGCN                  |                       | $R^2\uparrow$ | MAE (x1e-3)$\downarrow$ | RMSE (x1e-3)$\downarrow$ |
> | ------------------------- | --------------------- | ------------- | ------------------------ | ------------------------- |
> | [HR@10](mailto:HR@10)     | Performance Law       | **0.879**     | **4.665**                | **5.806**                 |
> |                           | Scaling Law           | 0.825         | 5.456                    | 6.978                     |
> |                           | Precision Scaling Law | 0.866         | 4.918                    | 6.116                     |
> | [NDCG@10](mailto:NDCG@10) | Performance Law       | **0.850**     | **5.423**                | **6.744**                 |
> |                           | Scaling Law           | 0.800         | 6.120                    | 7.802                     |
> |                           | Precision Scaling Law | 0.806         | 5.797                    | 7.223                     |
>
> | DiffuRec                  |                       | $R^2\uparrow$ | MAE (x1e-3)$\downarrow$ | RMSE (x1e-3)$\downarrow$ |
> | ------------------------- | --------------------- | ------------- | ------------------------ | ------------------------- |
> | [HR@10](mailto:HR@10)     | Performance Law       | **0.766**     | **34.933**               | **46.596**                |
> |                           | Scaling Law           | 0.608         | 50.277                   | 60.358                    |
> |                           | Precision Scaling Law | 0.652         | 46.882                   | 56.908                    |
> | [NDCG@10](mailto:NDCG@10) | Performance Law       | **0.748**     | **17.077**               | **20.707**                |
> |                           | Scaling Law           | 0.617         | 21.006                   | 25.505                    |
> |                           | Precision Scaling Law | 0.666         | 19.339                   | 23.818                    |
>
>
>
> > W2 The definitions of S^{real} (Real Entropy) and S^{\prime real} (the final measure of Real Entropy) are overly similar
>
>  Thanks for pointing out the presentation issue of this paragraph. In subsequent revisions, we will rename $(S^{real}$ to "Original Real Entropy" and modify $S^{\prime real}$ to $\overline{S^{real}}$ to enhance their distinctiveness.
>
>
> > W3 There are only the comparison between #Tokens·S^{\prime real} and #Tokens in In Figure 3. Including additional entropy-based data quality metrics in the comparison could further support the validity of the proposed Performance Law.
>
>   We have added experiments comparing the replacement of Real Entropy with Approximate Entropy [3], Shannon Entropy, and Kolmogorov Complexity (Implemented by compression rate)[4]. The results indicate that Real Entropy maintains the best fitting ability.
>
> |  $R^2$    | Real Entropy $S^{real}$| Token Only | Approximate Entropy | Shannon Entropy | Kolmogorov Complexity |
> | ---- | ------------ | ---------- | ------------------- | --------------- | --------------------- |
> | NG   | **0.9906**     | 0.8160     | 0.9825              | 0.7982          | 0.8955                |
> | HR   | **0.9913**     | 0.8259     | 0.9774              | 0.7998          | 0.8663                |
> | L    | **0.9881**   | 0.8776     | 0.9471              | 0.8617          | 0.9374                |
>
> \[3\]S.M. Pincus, Approximate entropy as a measure of system complexity., Proc. Natl. Acad. Sci. U.S.A. 88 (6) 2297-2301
>
> [4] Yin, M., Wu, C., Wang, Y., Wang, H., Guo, W., Wang, Y., Liu, Y., Tang, R., Lian, D., & Chen, E. (2024). Entropy Law: The Story Behind Data Compression and LLM
>
>
>  >Q1 Can you discuss the feasibility of the proposed Performance Law in other SR architectures / advanced generative techniques?
>
>  We appreciate your feedback! We have compared the fitting and predictive capabilities of the Performance Law against other baselines within the advanced generative frameworks of Mamba and DiffuRec (For DiffuRec, see reference W1 above). The Performance Law continues to demonstrate superiority.
>
> | Mamba                     |                       | $R^2\uparrow$ | MAE (x1e-3)$\downarrow$ | RMSE (x1e-3)$\downarrow$ |
> | ------------------------- | --------------------- | ------------- | ------------------------ | ------------------------- |
> | [HR@10](mailto:HR@10)     | Performance Law       | **0.818**         | **1.941**                    | **2.512**                     |
> |                           | Scaling Law           | 0.532         | 3.342                    | 4.021                     |
> |                           | Precision Scaling Law | 0.551         | 3.169                    | 3.805                     |
> | [NDCG@10](mailto:NDCG@10) | Performance Law       | **0.810**         | **0.398**                    | **0.530**                     |
> |                           | Scaling Law           | 0.663         | 0.553                    | 0.705                     |
> |                           | Precision Scaling Law | 0.663         | 0.532                    | 0.701                     |
>
>
>
>  >Q2 It would be beneficial to include additional entropy-based data quality metrics for comparison.
>
>   Thank you for your suggestions regarding entropy value comparisons. For detailed information, please refer to W3.

---

> > ### Comment · Reviewer_HQgL · 2025-08-06
> >
> > Thanks for your response! All of my concerns have been solved from your rebuttal. I truly hope that the authors can provide the codes of the other frameworks (GNN-based, DM-based and Mamba-based) and the details of the utilized entropy-based metrics in the final version. I will raise my score according to my reviews.

---

> > > ### Author Response · Authors · 2025-08-06
> > >
> > > Thank you for your decision, and we really appreciate your feedback. We've prepared the relevant content on GitHub, including the framework extensions (LightGCN, DiffuRec, Mamba4Rec) and entropy-based extensions (Approximate Entropy, Shannon Entropy, Kolmogorov Complexity).
> > >
> > > However, the anonymous GitHub repository is currently down, and since it includes links to external framework repositories, sharing it now might compromise the double-blind review policy.
> > >
> > > You will find these additions under the Performance_Law_Appendix_Framework directory after the repository is back online. We’ll also make sure to include all the links, results, and a clearer structure in the final version. Thanks again for your understanding and support!

---

> > > ### Author Response · Authors · 2025-08-07
> > >
> > > Since the anonymous repository is fixed, we have added the Framework Extensions (LightGCN, DiffuRec, Mamba4Rec) and the Entropy Extensions (Approximate Entropy, Shannon Entropy, Kolmogorov Complexity) mentioned in our original paper. You will find these additions under the Performance_Law_Appendix_Framework directory. Thank you very much again for your decision.

---

### Note · Authors · 2025-08-12

We are deeply grateful to all reviewers for their insightful comments and constructive feedback, which have been instrumental in refining our research. In this final remark, we aim to summarize our responses to the feedback provided and clarify any lingering questions about our work.

**Responses to Reviewers:**

1. **Generalizability Beyond SR (Reviewer HQgL, Reviewer 7fJJ, Reviewer UySy, Reviewer UaaA):**
   We acknowledge the reviewers' concerns regarding the generalizability of Performance Law beyond sequential recommendations. To counter this, we extended our experiments to include frameworks such as LightGCN for graph-based learning, Wukong for content-based recommendations, Diffusion for generative structure, and Mamba for longer sequential modeling, as emphasized in the rebuttal. These additions demonstrate the applicability of our approach beyond SR datasets, highlighting its versatility across different types of recommendation systems.
2. **Entropy Measures and Comparison (Reviewer HQgL, Reviewer UaaA):**
   The choice of Real Entropy over other measures such as Approximate Entropy, Shannon Entropy, and Kolmogorov Complexity was justified with empirical evidence showing superior fitting capability and robustness. The comparisons illustrate that our entropy measure captures data quality more adequately, aligning with our objective to optimize sequence data operations effectively.

Overall, all reviewers acknowledged that we have effectively addressed their concerns.


In conclusion, our work, Performance Law, provides a theoretically grounded yet practically feasible framework for enhancing the prediction accuracy and optimization of SR models under varying conditions. This significantly contributes to the scalability and robustness of recommendation systems in industrial settings, paving the way for future innovations in the field.

We sincerely thank the reviewers for their engagement and feedback, which greatly contributed to the refinement and depth of our work. We are committed to sharing our code and datasets publicly post-review, ensuring transparency and reproducibility, and fostering further exploration and development within the community. Again, we express appreciation and gratitude to the Area Chairs and reviewers for their efforts in reviewing this work, and we eagerly await the final decision on our submission.

---

### Decision · Program_Chairs · 2025-09-17

**Decision:**

Accept (poster)

**Comment:**

This paper introduces the Performance Law for sequential recommendation algorithms, enabling prediction of model performance for various settings. It also provides a quantitative framework for guiding the parameter optimization of future models. Extensive experiments show that the proposed approach has superior predictive accuracy compared to the tested baselines. It was found useful for optimization tasks as well.

The reviewers appreciated the use of Performance Law tailored to recommendation systems. While the experiments provided in the initial submission was found as lacking sufficient generality, the additional experiments along with other revisions provided in the rebuttal and discussions alleviated the concerns of the reviewers. Therefore, I feel that the paper can be a useful contribution at NeurIPS, and I encourage the authors to include the suggested revisions.